# Physics vs Distributions: Pareto Optimal Flow Matching with Physics Constraints

**Giacomo Baldan**
Technical University of Munich
Garching, Germany
`giacomo.baldan@tum.de`

**Qiang Liu**
Technical University of Munich
Garching, Germany
`qiang7.liu@tum.de`

**Alberto Guardone**
Politecnico di Milano
Milan, Italy
`alberto.guardone@polimi.it`

**Nils Thuerey**
Technical University of Munich
Garching, Germany
`nils.thuerey@tum.de`

## Abstract

Physics-constrained generative modeling aims to produce high-dimensional samples that are both physically consistent and distributionally accurate, a task that remains challenging due to often conflicting optimization objectives. Recent advances in flow matching and diffusion models have enabled efficient generative modeling, but integrating physical constraints often degrades generative fidelity or requires costly inference-time corrections. Our work is the first to recognize the trade-off between distributional and physical accuracy. Based on the insight of inherently conflicting objectives, we introduce *Physics-Based Flow Matching* (PBFM) a method that enforces physical constraints at training time using conflict-free gradient updates and unrolling to mitigate Jensen's gap. Our approach avoids manual loss balancing and enables simultaneous optimization of generative and physical objectives. As a consequence, physics constraints do not impede inference performance. We benchmark our method across three representative PDE benchmarks. PBFM achieves a Pareto-optimal trade-off, competitive inference speed, and generalizes to a wide range of physics-constrained generative tasks, providing a practical tool for scientific machine learning.
Code and datasets available at `https://github.com/tum-pbs/PBFM`.

## 1 Introduction

Partial differential equations (PDEs) provide the core mathematical framework for modeling the evolution of physical systems across space and time (Evans, 2010). However, discretizing PDEs often results in high-dimensional problems that are computationally prohibitive, especially for nonlinear or multiscale phenomena (Haber et al., 2018; Valencia et al., 2025). Recently, machine learning-based methods have emerged as efficient alternatives, enabling the approximation of PDE solutions with significantly reduced computational cost (Chen et al., 2021; Fresca et al., 2021; Baldan et al., 2021; Brunton & Kutz, 2024). Physics-informed neural networks (PINNs) (Raissi et al., 2019) embed PDE constraints directly into the training objective via automatic differentiation. However, PINNs yield a single deterministic solution, limiting their use in scenarios where stochasticity is crucial, such as uncertainty quantification (Roy & Oberkampf, 2011; Abdar et al., 2021; Liu & Thuerey, 2024). In contrast, generative models like denoising diffusion probabilistic models (DDPMs) (Ho et al., 2020; Nichol & Dhariwal, 2021), their implicit variants (DDIMs) (Song et al., 2022), and flow matching (Lipman et al., 2023) have shown strong performance in capturing complex data distributions across domains, including images, videos, audio, and graphs (Rombach et al., 2022; Ho et al., 2022; Kong et al., 2021; Chamberlain et al., 2021). Flow matching, in particular, offers a conceptually simple and computationally efficient framework, achieving high-fidelity sample generation with fewer function evaluations (Esser et al., 2024).

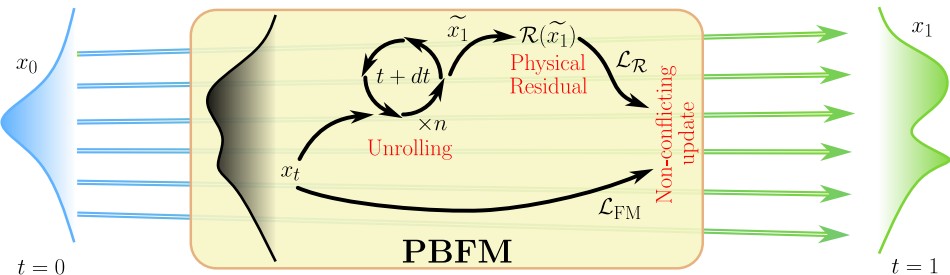

Figure 1: Overview of the proposed *Physics-Based Flow Matching* (PBFM) approach. During training, the sample $x_t$ at time $t$ is evolved to $t = 1$ over $n$ time steps to compute the residual $\mathcal{R}(\widetilde{x}_1)$. The flow matching loss $\mathcal{L}_{\text{FM}}$ and residual loss $\mathcal{L}_{\mathcal{R}}$ are combined in a conflict-free manner.

Despite recent progress, integrating physical constraints into generative models remains challenging. Enforcing such constraints during training requires the model to jointly optimize for generative fidelity and physical consistency, often resulting in conflicting gradients that can compromise either distributional accuracy or physical validity (Krishnapriyan et al., 2021). Achieving a balanced trade-off between these objectives is difficult, as improvements in one can degrade the other. In this work, we revisit physics-constrained generative modeling from a Pareto perspective, explicitly targeting solutions that balance physical correctness and generative performance; we systematically analyze the trade-offs involved.

Additionally, the so-called *Jensen's gap* exists: a fundamental discrepancy that arises when physical constraints, such as PDE residuals, are imposed on the posterior mean $\mathbb{E}[x_1|x_t]$ at intermediate noise levels, rather than directly on the final clean sample $x_1$ (Zhang & Zou, 2025). Alternatively, enforcing constraints only at inference time typically requires iterative sampling procedures to achieve physical consistency, which can be computationally expensive and undermine the efficiency of generative models. Moreover, enforcing nonlinear hard constraints necessitates differentiating through the physical residual, a process that is computationally expensive for high-dimensional and complex systems (Cheng et al., 2025).

We improve constrained flow matching with a series of established techniques such as unrolling, residual-based losses, stochastic sampling, and conflict-free gradient updates (Liu et al., 2025a), adapting them to address the unique challenges of physics-constrained generative modeling. Through the lens of the inherent physics-vs-distribution trade-off, we show that these modifications yield very substantial improvements over previous work, and enable training networks that yield distributional accuracy *and* satisfy physics constraints at the same time. Fig. 1 illustrates the overall training procedure and highlights the key components of PBFM method. Our main contributions are as follows. (**I**) We introduce a method that integrates physical constraints into flow matching, enabling the minimization of both PDE and algebraic residuals without manual balancing. (**II**) We show that unrolling during training effectively mitigates Jensen's gap, leading to lower residual errors and improved final predictions without increasing inference cost. (**III**) We analyze the role of Gaussian noise in flow matching under physical constraints, and (**IV**) we conduct a comprehensive comparison between deterministic and stochastic flow matching samplers, demonstrating advantages of the latter. A central advantage of the proposed approach is that it is very easy to implement in existing flow matching pipelines, and, as we will demonstrate below, consistently achieves substantial improvements in terms of distributional and physical accuracy.

## 2 RELATED WORK

Numerous deep learning methods have been developed to address complex problems in physics and engineering (Morton et al., 2018; Wang et al., 2020; Sanchez-Gonzalez et al., 2020; Thuerey et al., 2020). More recently, there has been growing interest in bringing the foundation model paradigm to scientific machine learning. Efforts such as PDEformer (Ye et al., 2024), Poseidon (Herde et al., 2024), Aurora (Bodnar et al., 2024), and Unisolver (Zhou et al., 2025a) aim to build models with broad generalization capabilities across diverse physical systems. In parallel, specialized transformer architectures tailored for PDEs have also been proposed, including OFormer (Li et al., 2023),

Transolver (Wu et al., 2024), Fengbo (Pepe et al., 2025), and PDE-Transformer (Holzschuh et al., 2025).

Focusing on diffusion models in the physics and engineering fields, representative applications include the generation and design of new molecules and drugs (Guo et al., 2024; Schneuing et al., 2024; Bose et al., 2024; Guastoni & Vinuesa, 2025), the simulation of particle trajectories in collider experiments (Mikuni et al., 2023), solving inverse PDE problems (Holzschuh et al., 2023), and the modeling of particle motion in turbulent flows (Li et al., 2024). Alongside these efforts, several works have sought to embed physical knowledge or constraints directly into the diffusion process to enhance model performance. For instance, Huang et al. (2024) presented an approach to solve PDEs from partial observations by filling in missing information using generative priors. Zhou et al. (2025b) proposed an approach that embeds priors into the diffusion process to satisfy energy and momentum conservation laws and PDE constraints. Rixner & Koutsourelakis (2021) formulated a probabilistic generative model enforcing physical constraints through virtual observables.

More aligned with our work, Shu et al. (2023) proposed an algorithm that conditions the diffusion process on the residual gradient both at training and inference time, with a focus on super-resolution and reconstruction tasks based on random measures of turbulent flows. Despite being one of the first works in this direction, they do not directly enforce the residual. Similarly, Jacobsen et al. (2025) introduced CoCoGen, a method that employs the governing PDEs during inference, while leaving the training procedure unchanged. The approach improves the physical residual but slows down inference, which is crucial in real applications. Other approaches that enforce physical constraints at inference time include FFM (Kerrigan et al., 2023), DiffusionPDE (Huang et al., 2024), D-Flow (Ben-Hamu et al., 2024), ECI (Cheng et al., 2025), and PCFM (Utkarsh et al., 2025). These methods require iterative sampling procedures to achieve physical consistency, which can be computationally expensive and can take more than $10\times$ longer than the standard diffusion process (Utkarsh et al., 2025). The last two methods are designed to strictly satisfy the constraints, but the ECI method is limited to simple non-overlapping constraints, while PCFM is applicable to arbitrary constraints but requires differentiating through the residual, which is computationally expensive. In contrast, Bastek et al. (2025) proposed physics-informed diffusion models (PIDM), which incorporate an additional loss term during training to minimize physical residuals. Zhang & Zou (2025) introduced physics-informed distillation of diffusion models (PIDDM), to potentially solve the Jensen's gap issue by distillation that is fine-tuned to minimize the physical residual. However, this approach requires training two separate models and, like Bastek et al. (2025) does not resolve the inherent conflict between the generative and physical objectives. Other recent methods that adopt fine-tuning to reduce the physics constraints are PIRF (Yuan et al., 2025) and PCFT (Tauberschmidt et al., 2025). We provide a comparison of our method with existing physics-constrained generative models in Table 1.

Table 1: Comparison between our method and other physics-constrained generative models.

| Method | Physics at Training | Hard Constraints | Gradient Free Inference | Complex Constraints | Balanced Hyperparams |
|---|---|---|---|---|---|
| FFM [34] | ✗ | ✓ | ✓ | ✗ | ✗ |
| ProbConserv [24] | ✗ | ✓ | ✓ | ✗ | ✗ |
| CoCoGen [32] | ✗ | ✗ | ✗ | ✓ | ✗ |
| DiffusionPDE [31] | ✗ | ✗ | ✗ | ✓ | ✗ |
| PIDM [9] | ✓ | ✗ | ✓ | ✓ | ✗ |
| D-Flow [10] | ✗ | ✗ | ✗ | ✓ | ✗ |
| ECI [16] | ✗ | ✓ | ✓ | ✗ | ✗ |
| PCFM [64] | ✗ | ✓ | ✗ | ✓ | ✗ |
| PIDDM [71] | ✓ | ✗ | ✗ | ✓ | ✗ |
| PBFM (ours) | ✓ | ✗ | ✓ | ✓ | ✓ |

## 3 METHODOLOGY

### 3.1 PRELIMINARIES

A general time-dependent PDE in $n$ spatial dimensions can be written as $\mathbf{u}_t(\mathbf{x}, t) = \mathcal{L}[\mathbf{u}(\mathbf{x}, t)] + \mathbf{f}(\mathbf{x}, t), \quad \mathbf{x} \in \Omega \subseteq \mathbb{R}^n$, where $\mathbf{u}(\mathbf{x}, t)$ is the solution field, $\mathcal{L}$ is a spatial differential operator, and $\mathbf{f}(\mathbf{x}, t)$ denotes external forcing. For numerical approximation, the domain $\Omega$ and its boundary

$\partial\Omega$ are discretized, and continuous operators are replaced by discrete analogues (Karniadakis & Sherwin, 2005).

Physics constraints can be enforced by minimizing a residual function, which is constructed drawing from a wide range of formulations. In generative modeling, the stochasticity of residuals can be categorized into three main types: (i) For *steady-state* PDEs, the solution distribution arises from uncertainties in the physical parameters, and the residual is computed directly from the governing equations, e.g., $\mathcal{R}(\mathcal{L}(\mathbf{u}) + \mathbf{f}) = 0$. (ii) For *time-dependent* PDEs, the solution evolves over time and residuals are formulated to enforce conservation laws (such as mass, momentum, or energy) across temporal snapshots. (iii) *Algebraic constraints* can be used to impose additional consistency between physical fields, further enforcing the underlying physics of the system.

### 3.2 Physics-based flow matching

Integrating physical constraints into generative models introduces an inherent conflict: optimizing for physical fidelity often degrades distributional accuracy, and vice versa. Existing diffusion-based approaches (Shu et al., 2023; Bastek et al., 2025) typically employ a weighted objective:

$$\arg\max_{\theta} \mathbb{E}_{x_1 \sim q(x_1)}[\log p_\theta(x_1)] + \mathbb{E}_{x_1 \sim p_\theta(x_1)}[\log q_{\mathcal{R}}(\hat{r} = 0 \mid x_1)] \tag{1}$$

where $\hat{r}$ are virtual observables (Rixner & Koutsourelakis, 2021) of the residual $\mathcal{R}(x_1)$. For flow matching, this reduces to (Bastek et al., 2025):

$$\mathcal{L} = w_{\text{FM}}\mathcal{L}_{\text{FM}} + w_{\mathcal{R}}\mathcal{L}_{\mathcal{R}} = w_{\text{FM}}\|u_t^\theta(x_t, t) - u_t(x_t)\|_2 + w_{\mathcal{R}}\|\mathcal{R}(x_1(x_t, t))\|_2, \tag{2}$$

where $w_{\text{FM}}$ and $w_{\mathcal{R}}$ are adjusted manually to address the potential conflicts between generative and physical losses. However, tuning these weights is challenging: increasing the residual term often harms generative quality, while prioritizing the generative term undermines physical consistency.

To address this, we leverage multi-task optimization techniques that resolve conflicting gradient directions. Specifically, we adopt the *ConFIG* method (Liu et al., 2025a) to compute conflict-free updates for physics-based flow matching:

$$\mathbf{g}_{\text{update}} = (\mathbf{g}_{\text{FM}}^\top \mathbf{g}_v + \mathbf{g}_{\mathcal{R}}^\top \mathbf{g}_v)\mathbf{g}_v \tag{3}$$

$$\mathbf{g}_v = \mathcal{U}\left[\mathcal{U}(\mathcal{O}(\mathbf{g}_{\text{FM}}, \mathbf{g}_{\mathcal{R}})) + \mathcal{U}(\mathcal{O}(\mathbf{g}_{\mathcal{R}}, \mathbf{g}_{\text{FM}}))\right] \tag{4}$$

where $\mathbf{g}_{\text{FM}}$ and $\mathbf{g}_{\mathcal{R}}$ are the gradients of $\mathcal{L}_{\text{FM}}$ and $\mathcal{L}_{\mathcal{R}}$, respectively; $\mathcal{O}(\mathbf{g}_1, \mathbf{g}_2) = \mathbf{g}_2 - \frac{\mathbf{g}_1^\top \mathbf{g}_2}{|\mathbf{g}_1|^2}\mathbf{g}_1$ is the orthogonality operator, and $\mathcal{U}(\mathbf{g}) = \mathbf{g}/|\mathbf{g}|$ normalizes to unit length. The resulting update direction $\mathbf{g}_{\text{update}}$ guarantees simultaneous descent on both objectives: $\mathbf{g}_{\text{update}}^\top \mathbf{g}_{\text{FM}} > 0$ and $\mathbf{g}_{\text{update}}^\top \mathbf{g}_{\mathcal{R}} > 0$. Figure 2 represents the gradient composition.

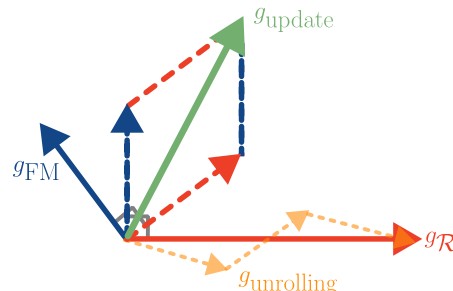

This approach adaptively aligns the gradients, eliminating the need for manual loss weighting and preventing one objective from dominating optimization. As a result, the resulting updates yield high distributional accuracy while still enforcing physical constraints, consistently outperforming fixed-weighted objectives across a wide range of weights (see Appendix F for an analysis).

Figure 2: PBFM method leverages ConFIG to combine the unit vectors of the orthogonal components and scales the result using the projection length of the FM and the physical residual $\mathcal{R}$ gradients.

### 3.3 Improving physical and distributional accuracy

A key requirement for generative models in scientific applications is the ability to produce samples with high physical accuracy, close to traditional numerical methods. In physics-constrained generative modeling, a central challenge is the so-called Jensen's gap which appears whenever a nonlinear map $f$ is applied to a random variable $Z$: in general $\mathbb{E}[f(Z)] \neq f(\mathbb{E}[Z])$. When physical constraints are imposed on the posterior mean $\mathbb{E}[x_1|x_t]$ at intermediate noise levels, rather than directly on the final clean sample $x_1$, a discrepancy arises that can degrade physical fidelity (Zhang & Zou,

2025). To mitigate this, we identify the reconstruction of the final, noise-free sample during training as crucial for accurately evaluating the physical residual. Specifically, we employ *unrolling*, i.e., integrating the intermediate sample at time $t$ forward to the final time $t = 1$ using multiple ODE steps. This process mitigates Jensen's gap by ensuring that the residual is evaluated on a more accurate prediction of $x_1$, rather than a single-step approximation. By unrolling over $n$ steps of size $(1 - t)/n$, the integration better approximates the true trajectory. Unrolling is applied via a curriculum, gradually increasing the number of steps during training. Additionally, we down-weight less accurate predictions near $t = 0$ by applying a scaling factor $t^p$ (with $p_{opt} = 1$) to the residual loss, further improving the learning signal. Linear scaling is also principled, aligning with the linear noise schedule in flow matching and ensuring consistency between residual weighting and FM dynamics. An investigation of different power laws is provided in Appendix E. The improved prediction of $x_1$ substantially enhances the evaluation of the physics residual in Eq. 2, yielding better optimization directions and directly mitigating Jensen's gap. This leads to lower residual errors and improved final predictions, without increasing inference cost. Nevertheless, unrolling increases memory consumption during training, as intermediate states must be stored for backpropagation. Table 6 details the additional training time and memory requirements for different numbers of unrolling steps.

Another important aspect in physics-based flow matching is the choice of $\sigma_{min}$, the amount of Gaussian noise added to training data adhering to the mixture of Gaussians theory. While computer vision tasks typically use $\sigma_{min} = 10^{-3}$ (Lipman et al., 2023; Esser et al., 2024), excessive noise perturbs physical residuals and degrades performance. The value of $\sigma_{min}$ thus influences the minimum achievable residual error. We provide an analysis of different noise levels in Table 5. A practical guideline is that adding Gaussian noise of scale $\sigma_{min}$ induces a residual MSE $\approx \sigma_{min}^2$ in a perfect reconstruction setting, reducing to $\sigma_{min} \lesssim \mathcal{R}_{min}$.

Inspired by natural image generation (Esser et al., 2024), we also sample the time variable $t$ from a logit-normal distribution (zero mean, unit standard deviation) during training, instead of a uniform distribution. This specifically targets regions where flow matching exhibits higher errors, typically around $t = 0.5$.

Algorithm 1 details the resulting training procedure, where the FM loss is computed at time $t$ and unrolling is used for accurate residual evaluation. The initial time is stored to weight the residual loss.

---

**Algorithm 1** Training procedure for PBFM

$n \leftarrow$ number of unrolling steps
$dt \leftarrow (1 - t)/n$             $\triangleright$ Compute $dt$ to reach the final state at $t = 1$
$\widetilde{t} \leftarrow t$             $\triangleright$ Keep the starting time $t$ to weight the residual loss
$u_t^\theta \leftarrow \text{model}(x_t, t)$           $\triangleright$ Flow matching velocity $u_t^\theta$ is not part of unrolling
$\widetilde{x}_1 \leftarrow x_t + dt \cdot u_t^\theta$
**for** $i = 1$, $i < n$ **do**             $\triangleright$ Improved state prediction via unrolling
    $\widetilde{t} = \widetilde{t} + dt$
    $\widetilde{u}_t^\theta \leftarrow \text{model}(\widetilde{x}_1, \widetilde{t})$
    $\widetilde{x}_1 \leftarrow \widetilde{x}_1 + dt \cdot \widetilde{u}_t^\theta$          $\triangleright$ Current fields $\widetilde{x}_1$ are updated until $t = 1$
**end for**
$\mathcal{R} \leftarrow \text{compute residual}(\widetilde{x}_1)$
$\mathcal{L}_\mathcal{R} \leftarrow \|t^p \cdot \mathcal{R}\|_2$           $\triangleright$ Residuals are weighted with $t^p$, $p_{opt} = 1$
$\mathcal{L}_{FM} \leftarrow \|u_t^\theta - u_t\|_2$
$\nabla_\theta \leftarrow \text{compute } \mathbf{g}_{update} \text{ via Eq. 3}$          $\triangleright$ Conflict-free update
AdamW optimizer step with $\nabla_\theta$

---

At inference time, the final sample is typically obtained by evolving the initial noise, $\mathcal{N}(0, I)$ with ODE integration. This represents a deterministic sampler in which all the stochasticity is embedded in the initial noise sample (Gao et al., 2025). In contrast, the sampling process in DDPM is stochastic (Ho et al., 2020). Given that both diffusion models and flow matching can be seen generative modeling variants under arbitrary Markov processes (Holderrieth et al., 2025), we explore the use of a stochastic sampler, similarly to what has been applied in ECI method (Cheng et al., 2025), within the physically based flow matching framework. The central idea is to evolve from time $t$ to $t = 1$ and then return to $t + dt$ using a different noise sample. This step backwards in time with newly generated noise increases the stochasticity in the sampling process and improves distributional accuracy. The resulting procedure is outlined in Algorithm 2.

## 4 EXPERIMENTAL SETUP

We evaluate the generative performance of our method, denoted with *Physics Based Flow Matching* (PBFM) in the following, on three benchmark problems. Across all experiments, we employ a diffusion transformer (DiT) backbone architecture (Peebles & Xie, 2023), with minor modifications detailed in Appendix G. For completeness, we also provide a comparison to the UNet from previous work (Ronneberger et al., 2015; Bastek et al., 2025), shown in Fig. 10. Although our method is agnostic to the type of physical residual, we organize the three benchmarks according to the residual categories described above: steady-state, transient, and analytic. For each case, the residual is computed using: finite differences for Darcy flow, FFT-based methods for Kolmogorov flow, and point-wise evaluation for dynamic stall. A complete description of datasets is provided in Appendix C.

**Darcy flow** We begin with the two-dimensional Darcy flow problem. The Darcy equation which models steady-state fluid flow through a porous medium. The solution comprises pressure $p$ and permeability $K$. We use the corresponding public dataset (Bastek, 2024), which contains 10k training and 1k validation samples, each of size $64 \times 64$. It is worth noting that, while common in literature (Zhu & Zabaras, 2018; Jacobsen et al., 2025), this dataset lacks conditioning inputs, making it less representative of real-world application scenarios. The residual directly corresponds to the governing PDE: $\mathcal{R} = \boldsymbol{\nabla} \cdot (K \boldsymbol{\nabla} p) + f = 0$

**Kolmogorov flow** The second benchmark is the two-dimensional Kolmogorov flow over a $128 \times 128$ spatial domain with periodic boundary conditions. The dataset is generated using a spectral solver and consists of velocity field snapshots for various Reynolds numbers in the range $[100; 500]$. The Reynolds number conditions the data generation, influencing the turbulence scales and flow complexity. The training set includes data for 32 Reynolds numbers with 1024 temporal snapshots each. The test set additionally includes 16 unseen Reynolds numbers. The data distribution reflects the temporal variation within each flow regime. The prediction target consists of the two velocity components, which are expected to satisfy the conservation of mass via: $\mathcal{R} = \boldsymbol{\nabla} \cdot \boldsymbol{U} = 0$

**Dynamic Stall** The final and most complex setup involves spatio-temporal fields over a pitching NACA0012 airfoil. This case captures the effects of *dynamic stall*, a complex, unsteady phenomenon in aerodynamics causing large-scale fluctuations and loads. As such, this physical model is highly relevant for real world cases such as helicopter and wind turbine blades, where dynamic stall plays a critical role. The solutions are obtained solving the compressible Navier-Stokes equations and the flow fields present shock waves and detailed flow phenomena that are challenging to capture. Each sample consists of $128 \times 128$ distributions of six quantities: absolute pressure, temperature, density, skin friction, and tangential velocity gradients (x and y). Conditioning is performed on four parameters that define the operating conditions and airfoil motion. The training set comprises 128 base configurations, each perturbed 32 times to model uncertainty, while the validation set includes 16 unseen configurations. Physical consistency across fields is enforced through two analytical, point-wise residual constraints: $\mathcal{R}_{\text{ig}}$ imposes the ideal gas law, while $\mathcal{R}_{\tau}$ minimizes the skin friction with Sutherland's law.

$$\mathcal{R}_{\text{ig}} = P - \rho R T \,, \qquad \mathcal{R}_{\tau} = \tau_w - \mu_0 \frac{T_0 + S}{T + S} \left(\frac{T}{T_0}\right)^{\frac{3}{2}} \sqrt{\left(\frac{\partial u}{\partial x}\right)^2_{n=0} + \left(\frac{\partial u}{\partial y}\right)^2_{n=0}} \qquad (5)$$

## 5 RESULTS

We evaluate our proposed framework, PBFM (Baldan et al., 2025a), in direct comparison with six representative baselines: flow matching with optimal transport FM-OT trained without physical constraints, PIDM-ME (Bastek et al., 2025), CoCoGen (Jacobsen et al., 2025), Diffusion-PDE (Huang et al., 2024), D-Flow (Ben-Hamu et al., 2024), and ECI (Cheng et al., 2025). ECI only enforces BCs but not the PDE due to the limitations of the method. For each method, we report physical residual error, distributional fidelity via Wasserstein distance and Jensen-Shannon divergence, number of function evaluations, and inference wall-clock time per sample. For the conditional benchmarks, Kolmogorov flow and dynamic stall, we additionally eval-

Table 2: Generative performance metrics for Darcy flow problem over 1024 samples. RE: physical Residual MSE, WD: Wassserstein Distance, JS: Jensen-Shannon divergence, NFE: Number of function evaluations, IT: Inference wall-clock time. **Best** and second best results.

| Metric | PBFM | FM-OT | CoCoGen† | PIDM† | DiffusionPDE | D-Flow‡ | ECI§ |
|---|---|---|---|---|---|---|---|
| RE | 0.838 | 4.159 | 1.320 | **0.022** | 3.388 | 2.286 | 3.045 |
| WD $\cdot 10^2$ | 0.138 | **0.059** | 0.249 | 3.103 | 0.089 | 0.147 | 2.892 |
| JS $\cdot 10^1$ | 0.256 | **0.131** | 0.360 | 3.179 | 0.139 | 0.237 | 2.818 |
| NFE | 20 | 20 | 100 | 100 | 20 | 20 | 20 |
| IT $[s]$ | 0.101 | **0.100** | 7.395 | 2.050 | 0.590 | 3.126 | 0.122 |

† This method uses the UNet architecture from Bastek et al. (2025).

‡ D-Flow method is unstable, samples are filtered using RE < 5 condition resulting in 888 valid ones.

§ The physical constraint is applied only to the BCs ($RE_{BC} \approx 0$) but cannot be applied to the non-linear PDE.

uate the mean and standard deviation of the predicted distributions for each conditioning input. Comprehensive sample visualizations and further analysis are provided in Appendix D.

## 5.1 DARCY FLOW

Unlike previous works that report only individual metrics, we present a comprehensive evaluation of Pareto optimality in terms of physical residual error and distributional accuracy. The latter is quantified by the Wasserstein distance (WD) (Villani, 2008; SciPy, 2025). Fig. 3 visualizes the trade-off for all compared methods. The FM-OT baseline yields a WD of 0.059 but incurs a high residual error of 4.159. DiffusionPDE achieves similar WD with marginally improved residuals. CoCoGen and D-Flow are strictly dominated by our approach, exhibiting both higher residuals and inferior distributional

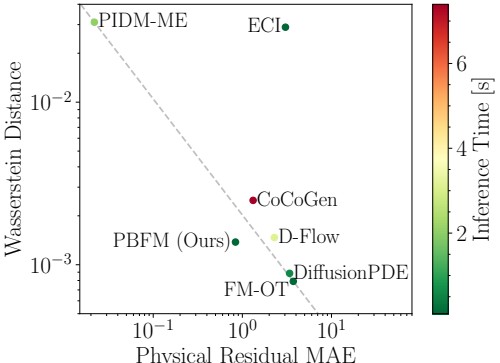

Figure 3: Pareto front of physical residual MAE vs. Wasserstein distance for Darcy flow for the proposed methods.

metrics. PIDM-ME attains the lowest residual (0.022), which comes at the expense of a substantially degraded distribution (WD 3.103), reflecting poor generative fidelity. In contrast, PBFM achieves a residual of 0.838 and WD of 0.138, demonstrating a more favorable balance between physical and generative objectives. A visual representation of the residual errors is reported in Fig. 5. The previous methods align along a negative-slope trend, highlighting the intrinsic trade-off between residual minimization and distributional accuracy. PBFM advances the Pareto front, reducing residuals without a considerable increase in WD. Comprehensive metrics are reported in Table 2, including Jensen-Shannon divergence (JS) (Lin, 1991) as a distributional metric in addition to WD, the number of function evaluations (NFE), and inference time (IT). PBFM maintains competitive inference speed (0.101 s), similarly to FM-OT, as the only difference is the use of the stochastic sampler.

The first panel of Fig. 4 illustrates two aspects. On the one hand, it shows how residual error evolves with the number of unrolling steps used during training and it demonstrates that unrolling effectively mitigates Jensen's gap, leading to lower residual errors. On the other hand, it highlights that only applying the ConFIG method during training offers a limited improvement. To further analyze the impact of unrolling, the second panel of Fig. 4 presents

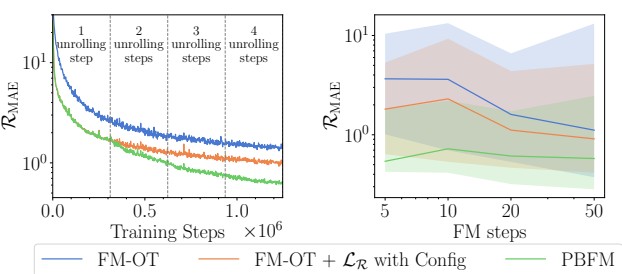

Figure 4: Darcy flow validation over 1024 samples using 20 FM steps. Residual MAE as a function of training steps (left), residual MAE as a function of FM steps (right).

the residual error as a function of the number of FM steps used during inference. The results confirm that increasing the number of FM steps consistently reduces the residual error across all methods. Notably, PBFM delivers the lowest residual error and it is less sensitive to the number of FM steps, indicating that it achieves high physical accuracy even with fewer function evaluations. This is particularly advantageous in practical applications where computational resources are limited.

To complete the analysis, we evaluate the impact of the stochastic sampler introduced in Algorithm 2. Table 3 reports the performance of PBFM with different thresholds $t^*$ for noise resampling during inference. Setting $t^* = 0.0$ corresponds to the deterministic sampler, while $t^* = 1.0$ implies resampling noise at every step. The results indicate that using the determin-

Table 3: Performance of stochastic sampler for Darcy flow problem over 1024 samples using 20 FM steps using PBFM. Noise resampling is enabled for $t < t^*$, details in Algorithm 2. $t^* = 0.0$ coincides with deterministic sampler. **Best** and second best results.

| Metric | $t^*$ | | | | | |
|---|---|---|---|---|---|---|
| | 0.0 | 0.2 | 0.4 | 0.6 | 0.8 | 1.0 |
| RE | 0.828 | 0.838 | 0.970 | 0.876 | 0.774 | **0.632** |
| WD $\cdot 10^2$ | 1.470 | **0.138** | 0.150 | 0.187 | 0.302 | 0.316 |
| JS $\cdot 10^1$ | 0.919 | **0.256** | 0.257 | 0.313 | 0.361 | 0.379 |

istic sampler yields comparable residual error to the stochastic approach, but with significantly worse distributional metrics. Focusing on values greater than 0, we observe that increasing $t^*$ leads to improved residual error, with the best performance at $t^* = 1.0$. However, this comes at the cost of degraded distributional accuracy, as both WD and JS increase. A balanced choice of $t^* = 0.2$ offers a good compromise, achieving low residuals while maintaining strong generative fidelity. Further results showing pressure and permeability samples and an additional UNet are reported in Fig. 9 and 10 of the appendix.

## 5.2 Kolmogorov flow

Kolmogorov flow presents a conditional generative modeling challenge, where the Reynolds number serves as the conditioning input and the physical residual is defined by the divergence of the velocity field, enforcing conservation of mass. Table 4 details the quantitative results. PBFM achieves the lowest physical residual er-

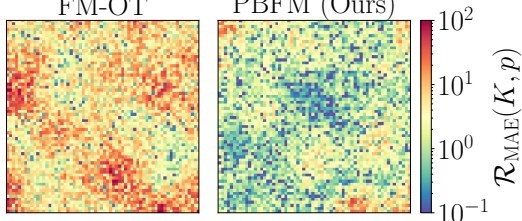

Figure 5: Physical residual of Darcy flow examples for the proposed method with 20 FM steps.

ror (1.362), outperforming all baselines. It also yields the best distributional metrics, with a low Wasserstein distance and Jensen-Shannon divergence, demonstrating excellent generative fidelity. The standard deviation error is slightly higher than FM-OT, but the overall distributional accuracy remains superior. Inference times for PBFM (98.97 ms) are competitive with FM-OT, while DiffusionPDE and D-Flow incur substantially higher computational costs due to FFT-based residual evaluation. D-Flow is unstable for Kolmogorov flow and does not produce valid samples despite the high computational cost.

Qualitative results are provided in Appendix Fig. 12 and 13, which show representative instantaneous velocity fields, divergence residuals, and the predicted mean and standard deviation across the test set.

## 5.3 Dynamic Stall

The dynamic stall case presents additional challenges, including two physical constraints of increased complexity and six predicted fields. The underlying phenomena are highly non-linear containing shock waves. Fig. 6 presents an example of predicted fields. Especially regions like the center exhibit strongly varying, small-scale shock waves that stochastically oscillate for perturbed operating conditions - a phenomenon that is important to capture for industrial applications (Baldan et al., 2025b).

Table 5: Metric comparison for different values of $\sigma_{\min}$ on the dynamic stall problem, 20 FM steps. **Best** and second best results.

| Metric | $\sigma_{\min}$ | | | |
|---|---|---|---|---|
| | 0.0 | $10^{-4}$ | $10^{-3}$ | $10^{-2}$ |
| RE $\cdot 10^6$ | **0.339** | 0.468 | 0.473 | 0.466 |
| WD $\cdot 10^4$ | **1.814** | 3.547 | 3.246 | 3.566 |
| JS $\cdot 10^2$ | **0.680** | 0.716 | 0.718 | 0.728 |
| MMSE $\cdot 10^5$ | 1.490 | 1.360 | 1.298 | **1.200** |
| SMSE $\cdot 10^5$ | 0.874 | 0.831 | **0.789** | 0.916 |

Table 4: Generative performance metrics for Kolmogorov flow and dynamic stall problems using 20 FM steps. RE: physical Residual MSE, WD: Wasserstein Distance, JS: Jensen-Shannon divergence, MMSE: MSE of the mean fields, SMSE: MSE of the standard deviation fields, NFE: Number of function evaluations, IT: Inference wall-clock time. **Best** and second best results.

| Dataset | Metric | PBFM | OT-FM | DiffusionPDE | D-Flow‡ | PCFM |
|---------|--------|------|-------|--------------|---------|------|
| Kolmogorov Flow | RE $\cdot 10^1$ | **1.362** | 2.314 | 1.930 | - | |
| | WD $\cdot 10^1$ | **1.222** | 2.124 | 3.698 | - | |
| | JS $\cdot 10^2$ | **7.440** | 12.49 | 19.39 | - | |
| | MMSE $\cdot 10^4$ | 4.455 | **4.188** | 5.669 | - | |
| | SMSE $\cdot 10^4$ | **2.484** | 2.574 | 21.82 | - | |
| | IT $[ms]$ | 98.97 | **98.75** | 267.8 | 6431 | |
| Dynamic Stall | RE $\cdot 10^6$ | 0.339 | 11.02 | 12.20 | 11.32 | **0.143** |
| | WD $\cdot 10^4$ | **1.814** | 2.707 | 2.509 | 3.484 | 4.013 |
| | JS $\cdot 10^2$ | **0.680** | 0.983 | 1.029 | 1.014 | 1.206 |
| | MMSE $\cdot 10^5$ | **1.490** | 2.791 | 2.626 | 2.507 | 5.669 |
| | SMSE $\cdot 10^5$ | **0.874** | 1.458 | 1.236 | 1.372 | 7.674 |
| | IT $[ms]$ | 60.47 | **59.75** | 171.7 | 138.9 | 3906 |

‡ D-Flow method is unstable for Kolmogorov Flow.

Table 4 summarizes the quantitative results. PBFM outperforms all baselines across nearly all metrics, achieving the lowest distributional errors and MSE for both the mean and standard-deviation fields, which indicates that PBFM effectively captures both first- and second-order statistics of the complex flow fields. PCFM achieves the lowest residual error, but this comes at the cost of worse distributional metrics and higher mean and standard-deviation errors. Notably, PCFM incurs a significantly higher inference time ($\approx 65\times$ that of PBFM) due to the iterative correction process. Inference times for PBFM are comparable to FM-OT. DiffusionPDE and D-Flow incur higher computational costs because evaluating their physical residuals is more complex, although the cost difference is limited since the residuals are pointwise.

To investigate the impact of different values of $\sigma_{\min}$ on the dynamic stall problem, which present the lowest residual errors and is more sensible to this phenomenon, Table 5 analyzes the results. Setting $\sigma_{\min} = 0$ yields the best distributional metrics and residual error underlines the importance of minimizing noise perturbation in this complex scenario. Increasing $\sigma_{\min}$ to $10^{-4}$ slightly degrades the metrics, while further increases to $10^{-3}$ and $10^{-2}$ lead to more pronounced declines in performance. This trend highlights the sensitivity of physical residuals to noise levels, emphasizing the need for careful tuning of $\sigma_{\min}$ in physics-constrained generative modeling. Nevertheless, with higher noise levels the model provides lower mean and standard deviation errors. A practical guideline is that adding Gaussian noise of scale $\sigma_{\min}$ induces a residual MSE $\approx \sigma_{\min}^2$

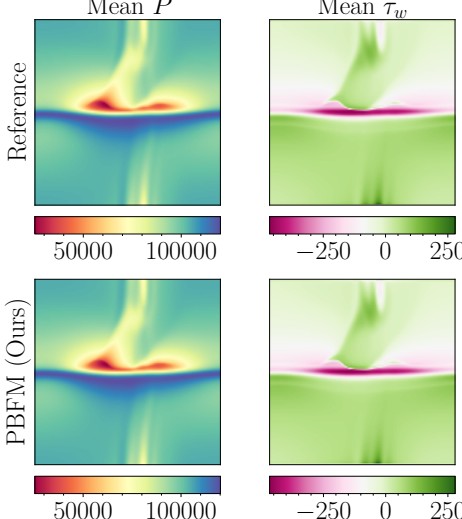

Figure 6: Dynamic stall, comparison of mean for PBFM framework computed over 128 samples using 20 FM steps.

in a perfect reconstruction setting. For dynamic stall, this directly suggests an upper bound: $\sigma_{\min}^2 \lesssim 10^{-7} \Rightarrow \sigma_{\min} \lesssim 3 \times 10^{-4}$. In contrast, Kolmogorov and Darcy benchmarks have more relaxed residual requirements. Importantly, this sensitivity is not unique to our method; it is general to flow matching formulations, as the noise level determines the scale of the denoising target and hence the attainable physical accuracy.

**Performance evaluation** We provide an overview of the training performance, focusing on the impact of the residual loss together with ConFIG, and the effect of unrolling. Table 6 reports the

wall-clock time, in seconds, for a single training iteration on an NVIDIA A100 64 GB GPU. To reduce training computation time and memory footprint, backpropagation can be computed only through the final unrolling step; this does not significantly affect model performance. In Appendix I we provide a detailed comparison of this variant against full backpropagation through all unrolling steps.

Table 6: Comparison of wall-clock time in seconds for one training iteration and memory usage in GB on an NVIDIA A100 64GB GPU for the proposed approaches. Batch size is 64 for all cases. Inference time is unchanged.

| | | FM | PBFM 1 step | PBFM 2 steps | PBFM 3 steps | PBFM 4 steps |
|---|---|---|---|---|---|---|
| Darcy | [s] | $4.28 \cdot 10^{-2}$ | $6.69 \cdot 10^{-2}$ | $9.78 \cdot 10^{-2}$ | $1.28 \cdot 10^{-1}$ | $1.59 \cdot 10^{-1}$ |
| | [GB] | 12.35 | 12.62 | 23.26 | 33.70 | 44.15 |
| Kolmogorov | [s] | $1.13 \cdot 10^{-1}$ | $1.94 \cdot 10^{-1}$ | $3.02 \cdot 10^{-1}$ | $4.10 \cdot 10^{-1}$ | $5.18 \cdot 10^{-1}$ |
| | [GB] | 4.08 | 4.52 | 6.91 | 9.56 | 12.18 |
| Dynamic | [s] | $4.29 \cdot 10^{-2}$ | $8.14 \cdot 10^{-2}$ | $1.18 \cdot 10^{-1}$ | $1.55 \cdot 10^{-1}$ | $1.90 \cdot 10^{-1}$ |
| | [GB] | 4.25 | 4.41 | 7.18 | 9.90 | 12.58 |

## 6 DISCUSSION AND CONCLUSIONS

**Limitations**  PBFM delivers substantial improvements in physical and distributional accuracy, but its main limitation is increased computational and memory cost during training. Table 6 details the wall-clock time and memory usage for different unrolling steps. Incorporating the residual loss and ConFIG requires an additional backward pass, increasing training time and memory usage. Unrolling over 4 steps further increases memory consumption (up to $3\times$) and training time (up to $2.5\times$). However, these overheads are restricted to training; the inference speed of the method remains unaffected.

**Discussion**  PBFM is designed to enable rapid prediction of complex physical systems, particularly in scenarios where conventional solvers are prohibitively slow. Table 10 presents inference times for the dynamic stall benchmark, already employed for actual helicopter blade simulations. While a single simulation using a CPU-based solver requires approximately 76 minutes, the trained PBFM model generates 128 samples in just 4 minutes on CPU and only 0.2 seconds on GPU. This substantial acceleration demonstrates the practical utility of PBFM for large-scale uncertainty quantification and surrogate modeling tasks in scientific and engineering workflows. Without the proposed modifications, a trained model would not be sufficiently accurate for real-world applications.

**Summary**  In this paper, we introduced PBFM, a generative framework designed to improve physical consistency while preserving the strengths of the flow matching approach for high-dimensional data generation. Our model provides straightforward improvements over standard FM to minimize physical residuals, arising from PDEs or algebraic constraints, in a conflict-free manner. We conduct extensive benchmarks across three representative physical systems, demonstrating the versatility and robustness of our approach. The incorporation of temporal unrolling plays a key role, enabling improved final state approximations and mitigating Jensen's gap by providing a more accurate prediction of the final sample. Additionally, our results highlight the benefits of stochastic sampling strategies, which outperform deterministic methods in cases involving complex target distributions. Beyond these specific benchmarks, the proposed framework generalizes to a wide class of PDE-constrained problems formulated via residual minimization. By combining the computational efficiency of flow matching with the interpretability and rigor of physics-based modeling, PBFM offers a powerful and flexible tool for surrogate modeling, uncertainty quantification, and simulation acceleration in physics and engineering applications.

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

## USE OF LARGE LANGUAGE MODELS

Large language models were used exclusively for text editing and correction. They were not involved in the generation of ideas, analyses, or substantive content.

## A    FLOW MATCHING

Flow matching has recently gained attention as a compelling alternative to diffusion models, offering a more direct and computationally efficient framework for generative modeling (Lipman et al., 2023). It enables high-quality sample generation with significantly fewer function evaluations (Esser et al., 2024). Given a known source distribution $p$ and an unknown target distribution $q$, flow matching learns a vector field $u_t^\theta$, parameterized by a neural network, that generates a probability path $p_t$ interpolating from $p_0 = p$ to $p_1 = q$ (Lipman et al., 2024). The learning objective is defined as:

$$\mathcal{L}_{\text{FM}}(\theta) = \mathbb{E}_{t \sim \mathcal{U}[0,1],\, x \sim p_t} \|u_t^\theta(x) - u_t(x)\|_2$$

where $\theta$ denotes the model parameters. While multiple formulations exist for the target velocity field $u_t$, a particularly simple and effective one leverages optimal transport (OT) (Tong et al., 2024). In this setting, samples from the base distribution $p_0 = \mathcal{N}(0, I)$ are linearly transported to $p_1$ via the conditional flow:

$$u_t(x \mid x_1) = \frac{x_1 - (1 - \sigma_{\min})x}{1 - (1 - \sigma_{\min})t},$$

with $\sigma_{\min} \sim 10^{-3}$, and the corresponding interpolant:

$$\psi_t(x) = (1 - (1 - \sigma_{\min})t)x + tx_1.$$

This yields a straight-line conditional flow with a time-independent vector field. Sampling from the trained model requires integrating the learned field over time:

$$x_1 = \int_0^1 u_t^\theta(x_t)\, dt,$$

typically using numerical ODE solvers such as Euler's method. Although the true OT vector field is constant, the learned approximation typically is not, and integration quality still depends on the time discretization.

## B    ABLATION WITH A TOY PROBLEM

To provide an intuition for the proposed improvements, we consider an ablation with a toy problem where the neural network outputs are the $x$ and $y$ coordinates of points on a circumference subject to the physical constraint of unit radius. Fig. 7 shows the resulting point distribution along with the corresponding absolute physical error through mean estimation of the final sample. We compare with diffusion models *DM*, as a DDPM representative (Song et al., 2022), and *PIDM-ME*, the best performing physics-informed algorithm (Bastek et al., 2025), and a variant of the flow matching approach that uses Config without unrolling. It is apparent that each extension yields a noticeable gain, and the final PBFM model exhibits a residual error that is 61.8 and 27.5 times lower than the DM and PIDM-ME baselines.

## C    DATASET GENERATION AND RESIDUAL COMPUTATION

We provide a detailed description of the datasets used and generated, as well as the method employed to compute residuals during training.

### C.1    DARCY FLOW

We use the dataset introduced by Bastek (2024) to enable direct comparison with their results. For completeness, we briefly summarize the key characteristics of the dataset. The underlying physical model is governed by the steady-state Darcy equations, which describe fluid flow through a porous medium:

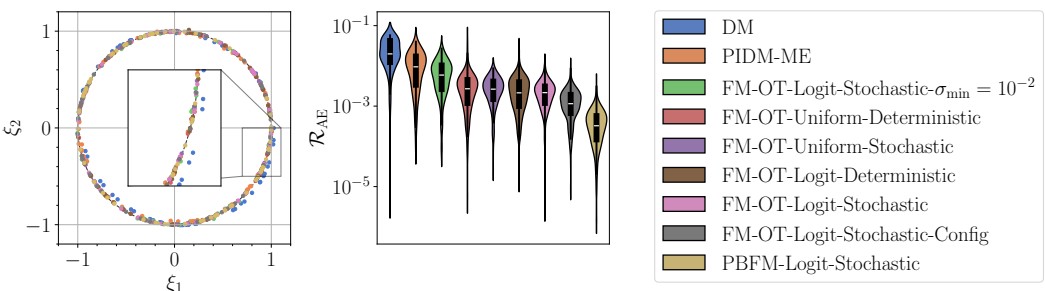

Figure 7: Point distribution and absolute error of the physical residual (circle radius squared) for SOTA reference DM, PIDM-ME (Bastek et al., 2025), and all proposed approaches.

$$\boldsymbol{u}(\boldsymbol{x}) = -K(\boldsymbol{x})\nabla p(\boldsymbol{x}), \quad \boldsymbol{x} \in \Omega$$
$$\nabla \cdot \boldsymbol{u}(\boldsymbol{x}) = f(\boldsymbol{x}), \quad \boldsymbol{x} \in \Omega$$
$$\boldsymbol{u}_{\hat{n}(\boldsymbol{x})} = 0, \quad \boldsymbol{x} \in \partial\Omega$$
$$\int_\Omega p(\boldsymbol{x})\, d\boldsymbol{x} = 0$$

Here, $\hat{n}(\boldsymbol{x})$ denotes the unit outward normal on the boundary $\partial\Omega$. The source term $f(\boldsymbol{x})$ is defined as:

$$f(\boldsymbol{x}) = \begin{cases} r & \text{if } \left|x_i - \frac{1}{2}w\right| \le \frac{1}{2}w \\ -r & \text{if } \left|x_i - 1 + \frac{1}{2}w\right| \le \frac{1}{2}w \\ 0 & \text{otherwise} \end{cases}$$

with $r = 10$ and $w = 0.125$. The permeability field $K(\boldsymbol{x})$ is modeled as $K(\boldsymbol{x}) = \exp(G(\boldsymbol{x}))$, where $G(\boldsymbol{x})$ is a Gaussian random field. The pressure field is generated by solving a least-squares problem based on a 64-term truncated Karhunen-Loève expansion, following the approach of Jacobsen et al. (2025).

To ensure consistency and avoid introducing numerical discrepancies, we adopt the same procedure for computing residuals during training as was used during dataset generation. Specifically, we employ identical finite difference stencils implemented via 2D convolutional layers, and apply the same reconstruction method for the forcing term $f$. The pressure field is also normalized by removing the integral contribution.

## C.2 KOLMOGOROV FLOW

We generate two distinct datasets (training and validation) for the Kolmogorov flow problem with Reynolds numbers in the range $[100, 500]$, using a spatial resolution of $128 \times 128$. The simulation is based on the vorticity–stream function formulation. The velocity field is obtained from the computed vorticity through the stream function, while the pressure field is derived by solving the pressure Poisson equation. We employ TorchFSM (Liu et al., 2025b) to perform GPU-accelerated flow simulations using the Fourier spectral method. The training dataset includes 32 different flow conditions sampled via a Halton sequence (Kocis & Whiten, 1997), while the validation dataset contains 16 conditions. For each condition, 256 simulations are conducted with slightly perturbed initial states. Simulations are run for 10 000 time steps to reach a statistically steady state, followed by data sampling every 4 000 time steps. With a time step size of $dt = 1/\mathrm{Re}$, this yields 1 024 snapshots per condition.

To ensure consistency, the divergence of the velocity fields is computed using the same numerical scheme as the spectral solver employed for dataset generation.

## C.3 DYNAMIC STALL

The dynamic stall datasets used for training and validation are generated by solving the unsteady, compressible, two-dimensional RANS equations around a sinusoidally pitching NACA0012 airfoil. An O-grid mesh is utilized, featuring 512 nodes along the airfoil surface and 128 nodes in the normal direction. Simulations are performed using ANSYS Fluent 2024R2 (ANSYS, 2024). The governing equations are discretized using a second-order upwind scheme for spatial accuracy and a second-order implicit scheme for time integration. Gradient reconstruction employs a least-squares cell-based method, while fluxes are computed using the Rhie-Chow momentum interpolation. Pressure-velocity coupling is handled through a coupled solver. Airfoil pitching is modeled by prescribing a rigid-body motion to the entire mesh, defined by the angular velocity function $\dot{\alpha}(t) = \omega\,\alpha_s\,\cos(\omega t)$. The mean angle of attack, $\alpha_0$, is applied at the start of each simulation by rotating the mesh accordingly. To close the RANS equations, the SST turbulence model with an intermittency transport equation is employed. Each oscillation cycle is resolved using 2 048 time steps, and simulations are run until periodic convergence is achieved (Baldan & Guardone, 2024; 2025).

Table 7: Range of conditioning inputs that define the operating conditions of the pitching airfoil.

| Variable | Min | Max |
|---|---|---|
| Mach | 0.3 | 0.5 |
| $\alpha_0$ | 5° | 10° |
| $\alpha_s$ | 5° | 10° |
| $k$ | 0.05 | 0.1 |

The design space is defined as a four-dimensional hypercube, with each axis corresponding to a conditioning input for the neural network. These inputs include the free-stream Mach number, the mean angle of attack $\alpha_0$, the pitching amplitude $\alpha_s$, and the reduced frequency $k = \omega c/2V_\infty$. The ranges for each variable are provided in Table 7. Training and testing datasets are constructed using Halton sequences. The hypercube is sampled with 128 points for training and 16 points for testing. Each sampled point represents a nominal operating condition. Each nominal condition is perturbed as follows:

$$x_{\text{perturbed}} = (1 + \mathcal{N}(0, 0.02))\ x_{\text{nominal}}$$

where $\mathcal{N}(0, 0.02)$ denotes a Gaussian noise term with zero mean and standard deviation 0.02. This results in 32 perturbed variations per nominal condition, yielding a total of $128 \times 32$ simulations for training and $16 \times 32$ for testing. To reduce computational costs, all simulation fields are downsampled to a resolution of $128 \times 128$. The saved quantities include fields of absolute pressure, density, temperature, signed skin friction, and tangential velocity gradients across the airfoil surface over a full pitching cycle. Fig. 8 illustrates a spatio-temporal representation of the wall shear stress ($\tau_w$) across one complete pitching cycle, highlighting the pitch-up and pitch-down phases and demonstrating the structured mapping of surface data over time.

## D ADDITIONAL SAMPLES AND ANALYSIS

We conclude the analysis of the proposed test cases by presenting additional comparisons and complete field prediction examples.

### D.1 DARCY FLOW

Fig. 9 shows the pressure and permeability fields for the FM-OT and PBFM methods under analysis. To ensure a fair comparison of the residuals, all fields have similar variable ranges, since higher magnitudes can lead to disproportionately large residual values. This choice is further justified by the residual plots, which exhibit higher absolute errors in regions with large permeability. Despite local differences, a clear difference emerges: the FM baseline presents residual peak values around 100, while PBFM method peak residuals are reduced by approximately an order of magnitude.

To conclude our analysis of the Darcy flow, Fig. 10 compares the DiT architecture with the UNet used by Bastek et al. (2025). This comparison corresponds to the FM setup without residual loss. In

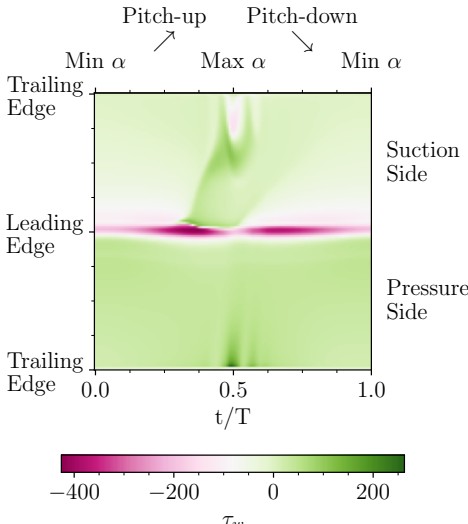

Figure 8: Example of a spatio-temporal contour of the post-processed $\tau_w$ distribution over an entire pitching cycle.

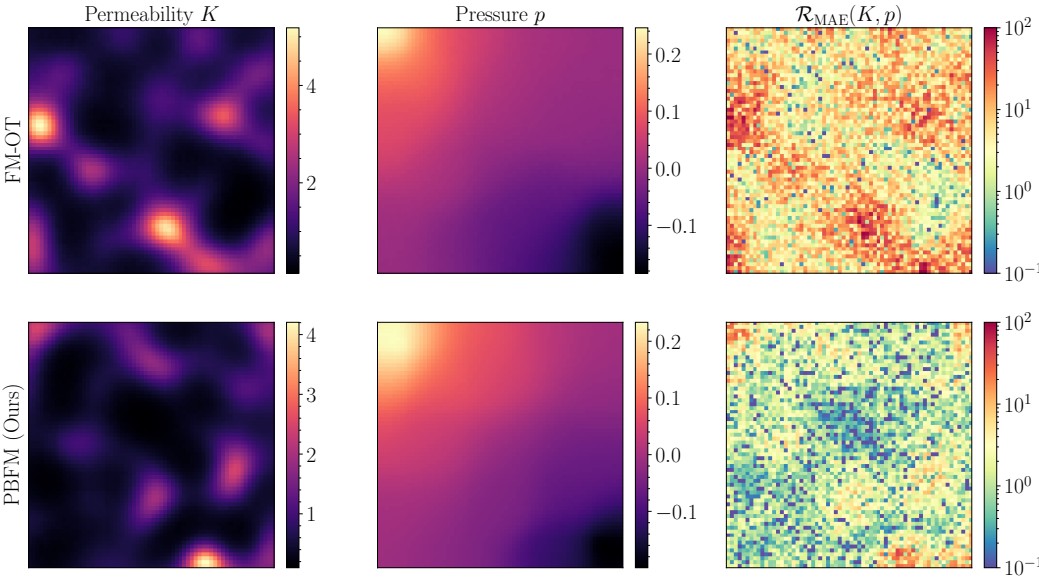

Figure 9: Darcy flow examples of pressure and permeability fields along with the physical residual for the proposed approaches.

the first panel, which reports the FM-OT loss during training, we observe that the UNet architecture reproduces the overfitting behavior noted by Bastek et al. (2025). In contrast, the DiT architecture exhibits a stable and monotonic reduction in training loss. The second panel shows the residual error, which remains comparable between the two models overall, with the exception of slight discrepancies at one function evaluation. Finally, in the third panel, the Wasserstein distance reveals a clear gap in performance: the UNet consistently incurs higher errors, with the distance being approximately twice as large for pressure and up to four times larger for permeability, underscoring the superior distributional accuracy of the DiT model. Additional samples produced by our method are shown in Fig. 11.

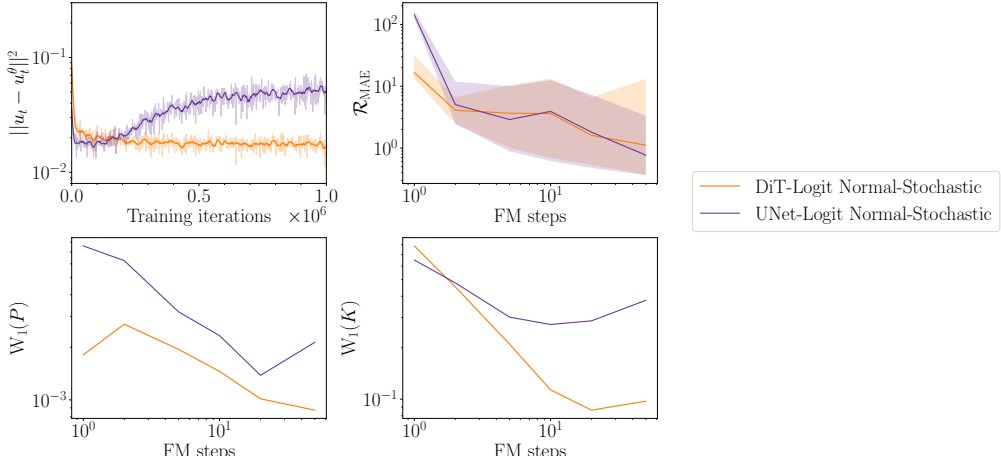

Figure 10: Comparison of UNet (Bastek et al., 2025) and DiT architectures for Darcy flow. The physical residual (error bars refer to min-max values within the validation dataset samples) and Wasserstein distance, as a function of FM steps, are computed over 1024 samples.

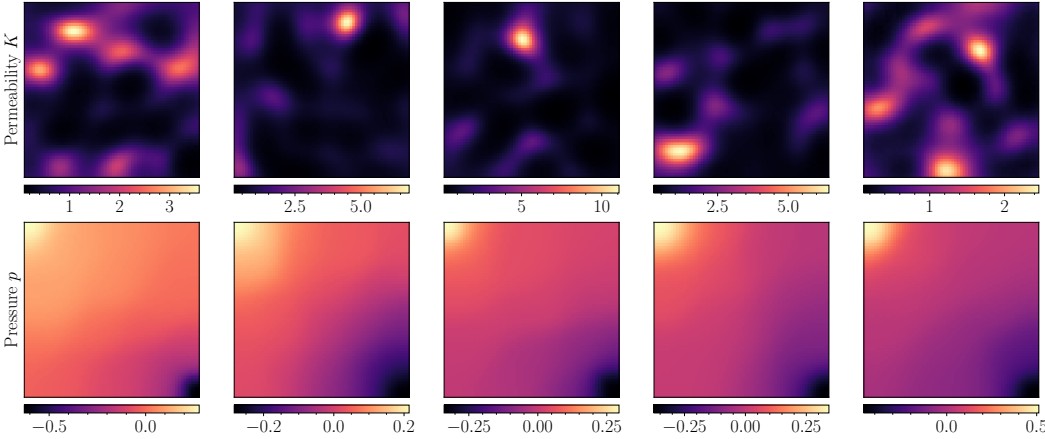

Figure 11: Representative Darcy flow field samples generated with PBFM using 20 FM steps. The samples illustrate the diversity in flow behavior, particularly highlighting variability in the range extrema.

### D.1.1 NON-DIMENSIONAL SCALING

Even though the role of scaling is well recognized in machine learning, its impact on *physical residuals* and *distributional performance* remains an important aspect to examine. We study this effect in the context of the Darcy flow problem by comparing two scaling strategies. In the first approach, each variable is normalized to have zero mean, yielding the following scaling values: $p_{\text{mean}} = 0.0$, $p_{\text{std}} = 0.576$, $k_{\text{mean}} = 1.386$, $k_{\text{std}} = 10.64$. The corresponding results are reported in Table 2. The second approach, denoted as the *Scale* version, rescales the pressure $p$ to the range $[-1, 1]$ and the permeability $k$ to $[0, 1]$. The results for this setting are shown in Table 8. Across all methods, this alternative scaling consistently degrades performance, leading to higher physical residual errors and poorer distributional metrics.

### D.2 KOLMOGOROV FLOW

Fig. 12 illustrates example predictions from the FM-OT, and PBFM models, along with their corresponding physical residuals, representing the divergence of the predicted fields. The baseline FM

Table 8: Generative performance metrics for Darcy flow problem over 1024 samples. RE: physical Residual MSE, WD: Wassserstein Distance, JS: Jensen-Shannon divergence, NFE: Number of function evaluations, IT: Inference wall-clock time. **Best** and second best results.

| Dataset | Metric | PBFM Scale | FM-OT Scale | DiffusionPDE Scale | D-Flow Scale‡ |
|---------|--------|-----------|-------------|--------------------|--------------| 
| | RE | **2.260** | 5.551 | 4.307 | 2.417 |
| | WD $\cdot 10^2$ | 0.185 | 0.076 | **0.075** | 0.437 |
| Darcy | JS $\cdot 10^1$ | 0.218 | 0.152 | **0.142** | 0.353 |
| | NFE | 20 | 20 | 20 | 20 |
| | IT $[s]$ | 0.101 | **0.100** | 0.590 | 3.126 |

‡ D-Flow method is unstable, samples are filtered using RE $< 5$ condition resulting in 403 valid ones.

exhibits a residual MSE on the order of $10^{-1}$ across most of the domain, whereas the unrolled framework significantly reduces the error, dropping below $10^{-3}$ in large regions of the field, despite some localized areas with higher residuals.

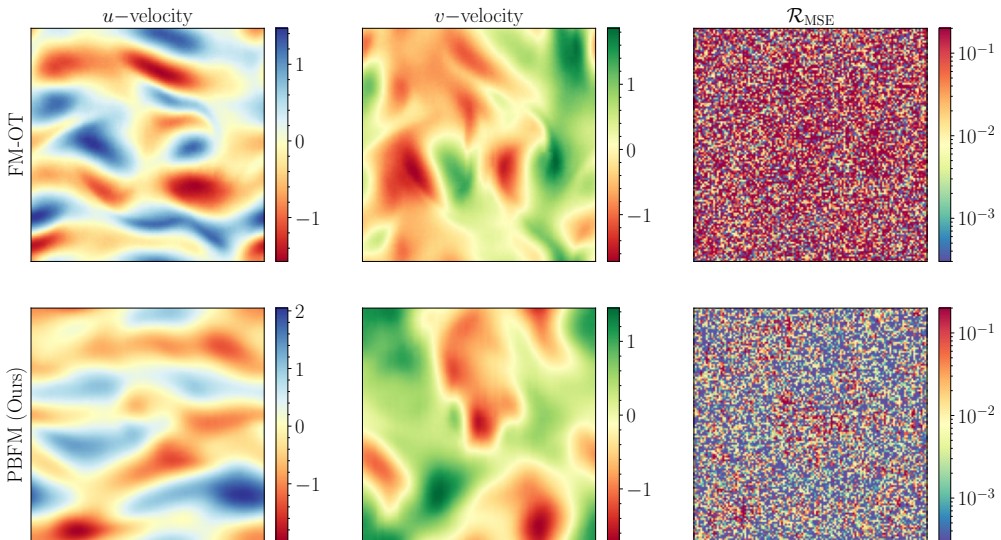

Figure 12: Kolmogorov flow example of $u-$ and $v-$velocity fields and physical residual, divergence, MSE for the proposed approaches using 20 FM time steps.

Furthermore, Fig. 13 shows the mean and standard deviation of the predicted fields for FM-OT and PBFM models, alongside the reference data. The frameworks closely match the reference mean, with the unrolled version providing a smoother, less oscillatory solution compared to FM-OT. Regarding the standard deviation, none of the models fully capture the reference distribution, particularly missing the peak value of approximately 0.95, although their performance remains broadly comparable.

Additional samples produced by PBFM method are shown in Fig. 14.

## D.3 DYNAMIC STALL

For the dynamic stall case, Fig. 15 presents the mean and standard deviation of the predicted fields obtained using PBFM. The predictions show strong agreement with the reference data, both in terms of overall distribution as well as capturing the extrema of each variable. The only notable deviation appears in the standard deviation of the skin friction, which slightly exceeds the corresponding reference values.

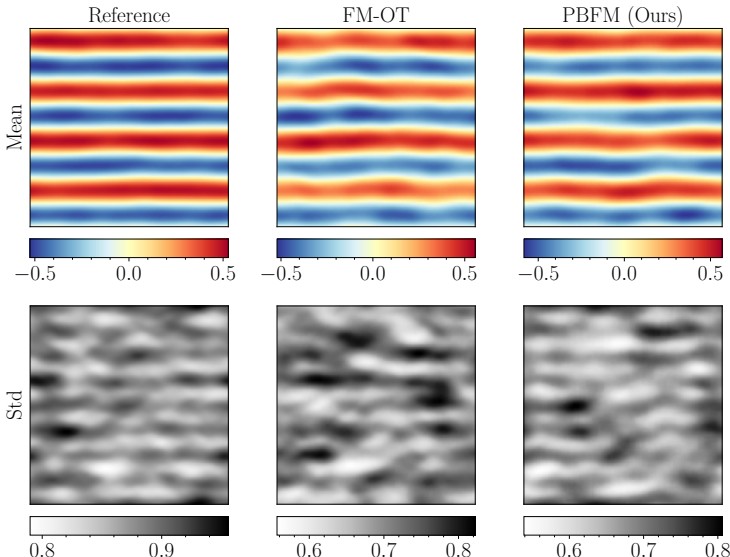

Figure 13: Example of mean and standard deviation of Kolmogorov flow computed over 20 FM steps with 128 samples for the proposed approaches.

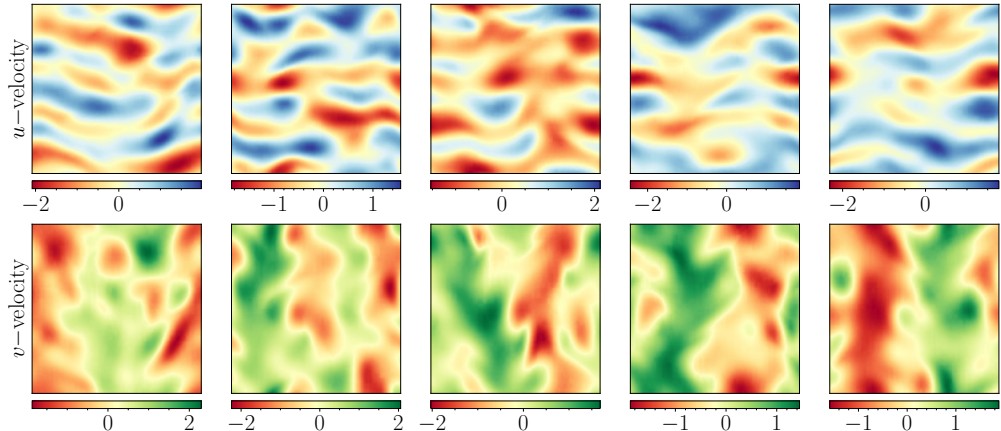

Figure 14: Example of Kolmogorov flow field samples generated with PBFM using 20 FM steps.

Fig. 16 compares the performance of the proposed models as a function of the number of FM integration steps. The first panel focuses on the physical residual MSE, indicating that the best performance is achieved with 10 FM steps, with PBFM consistently delivering a one order of magnitude reduction compared to the baseline. The second panel reports the Wasserstein distance, showing that PBFM also narrows the gap with the reference data distribution, reaching its optimal value at 20 FM steps. Finally, the MSE of the mean and standard deviation reveals that the proposed methods preserve the already strong performance of the baseline, introducing only minor deviations.

Additional qualitative samples produced by our trained PBFM method are shown in Fig. 17. They highlight the wide range of physical behavior modeled by this case: from small oscillation attached flow at the top, to deep dynamic stall cases with shock wave formation at the bottom.

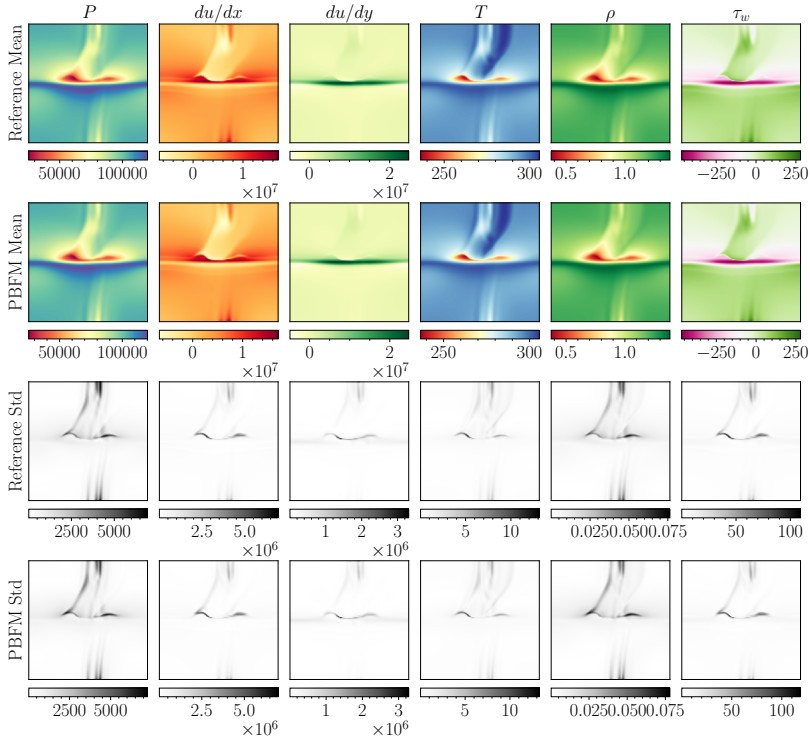

Figure 15: Example of mean and standard deviation of dynamic stall problem computed over 128 samples with 20 FM steps.

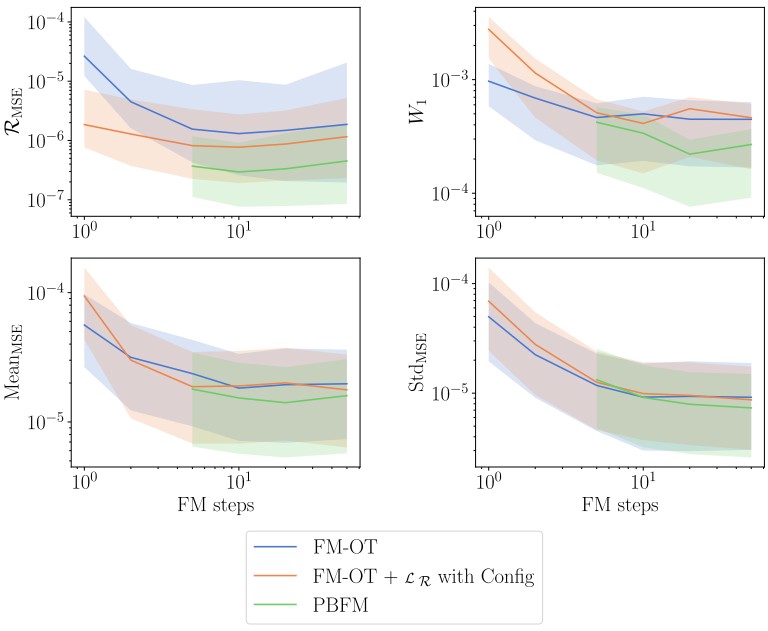

Figure 16: Comparison of the proposed approaches for physical residual and Wasserstein distance, mean and standard deviation MSE as a function of FM steps for dynamic stall case. Error bars refer to the different conditioning values within the validation dataset.

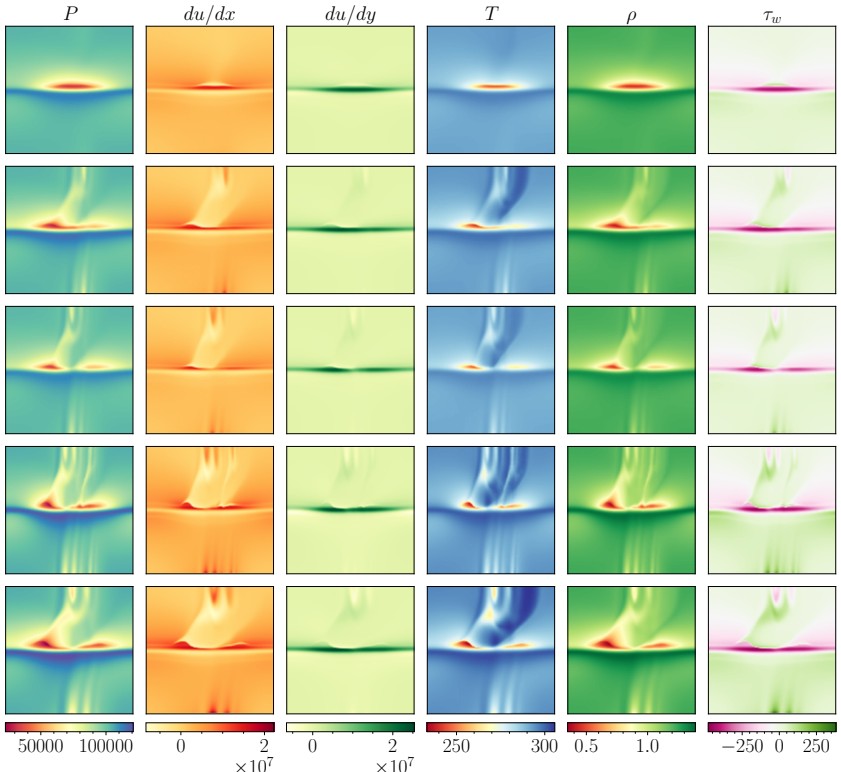

Figure 17: Example of dynamic stall flow field samples generated with PBFM using 20 FM steps. The outputs vary substantially depending on the conditioning inputs, illustrating the model's sensitivity to different flow scenarios.

## E    RESIDUAL LOSS SCALING LAWS

During training, the residual of each sample is computed starting from a given time $t$ and evolved forward until $t = 1$. Trajectories initialized at times closer to $t = 0$ tend to exhibit larger errors, particularly when only a single integration step is used. To mitigate this effect, we introduce a weighting scheme based on a power law $t^p$, where the residual loss is scaled according to the starting time $t$. We investigate the impact of different power exponents $p$, focusing on the most challenging case of dynamic stall.

Fig. 18 presents a comparison of the MSE for physical residuals, as well as the mean and standard deviation, for both the FM-OT with Config and PBFM frameworks. The results show that unrolling helps regularize the error, producing a monotonic increase in error as a function of the power $p$, and also reduces sensitivity to the choice of $p$ in the range $[1, 4]$. Notably, both frameworks achieve optimal performance when residuals are scaled linearly with time, $p = 1$. In contrast, using no scaling, $p = 0$, results in significantly higher errors, underscoring the importance of appropriately weighting residuals based on the starting time.

## F    CONFLICT-FREE UPDATES AND WEIGHTED LOSS TERMS

The introduction of the second loss term associated with the physical residual minimization transforms the framework into a multi-task learning problem. A seemingly attractive approach is to combine the two losses using a fixed weighting hyperparameter. However, this naive strategy requires manual tuning of the relative weight and often leads to suboptimal performance. In particular, optimization may get stuck in a local minimum of one loss due to conflicts between the tasks. To address this, we adopt the conflict-free updates (Liu et al., 2025a), which mitigate gradient conflicts by

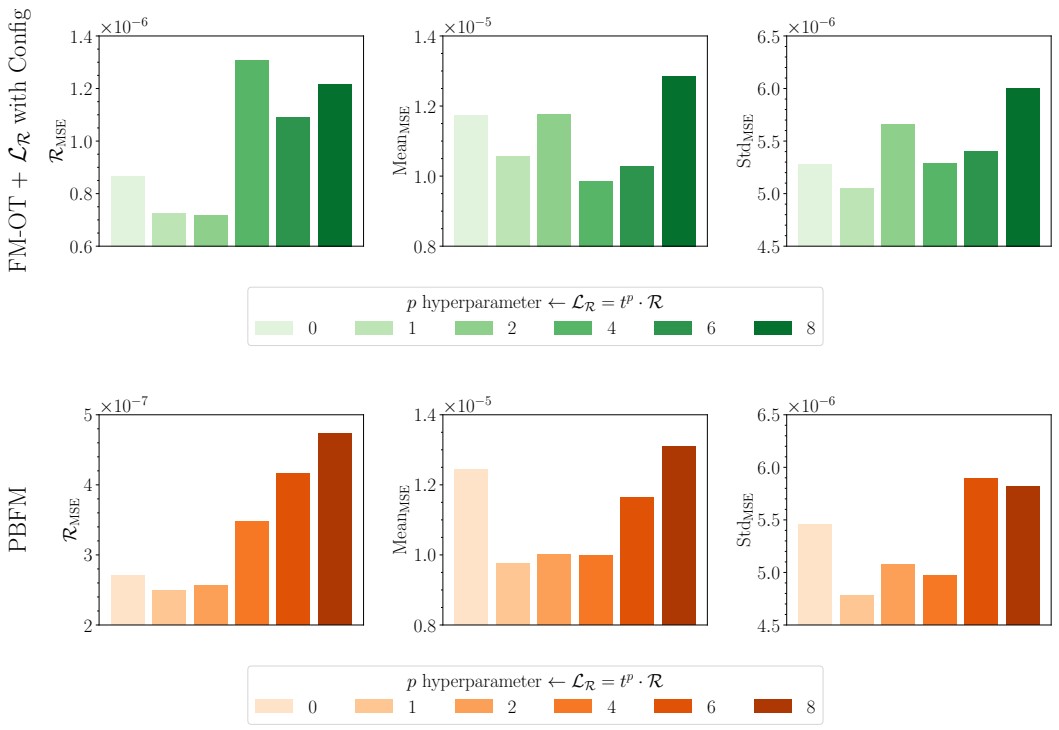

Figure 18: Comparison of physical residual, mean and standard deviation MSE for dynamic stall under various power-law scalings in the residual loss. Unrolling reduces sensitivity to the scaling exponent.

computing a non-conflicting optimization direction through the inverse of the loss-specific gradient covariance matrix. Furthermore, this approach has the potential to yield improved solutions.

We evaluate the proposed setup on the dynamic stall case, the most challenging scenario considered, comparing ConFIG to models trained with various fixed loss weights. Fig. 19 reports the MSE for the physical residual, the predicted mean, and the standard deviation across all configurations. ConFIG consistently outperforms the fixed-weight approaches, achieving the lowest error in each individual metric and delivering the best overall performance.

To further assess the effectiveness of ConFIG, we quantified gradient conflicts via pairwise cosine similarity between physical and FM losses. For the best model using 4-step unrolling with ConFIG, the average conflict is 6.93%, whereas a model with fixed residual scaling at 500 exhibits a much higher average conflict of 19.98%.

## G ARCHITECTURE AND TRAINING DETAILS

The framework is implemented in PyTorch v2.5.1, employing Distributed Data Parallel (DDP) for scalable training. All experiments are trained using the AdamW optimizer with weight decay set to 0, $\beta_1 = 0.5$, $\beta_2 = 0.999$, and a fixed learning rate. To avoid learning rate adjustments across different hardware setups, we maintain a constant global batch size, independent of the number of GPUs. An Exponential Moving Average (EMA) of the model parameters is maintained throughout training, with a decay rate of 0.999, and is used during sampling.

We adopt the Diffusion Transformer (DiT) architecture proposed by Peebles & Xie (2023) as the backbone for our flow matching model, incorporating minor modifications. The model is conditioned via adaptive layer normalization (adaLN-Zero) blocks, which replace the standard normalization layers. The scale and shift parameters in these blocks are derived from the sum of the embedding vectors for the time step $t$, used in the flow matching process, and the conditioning signal

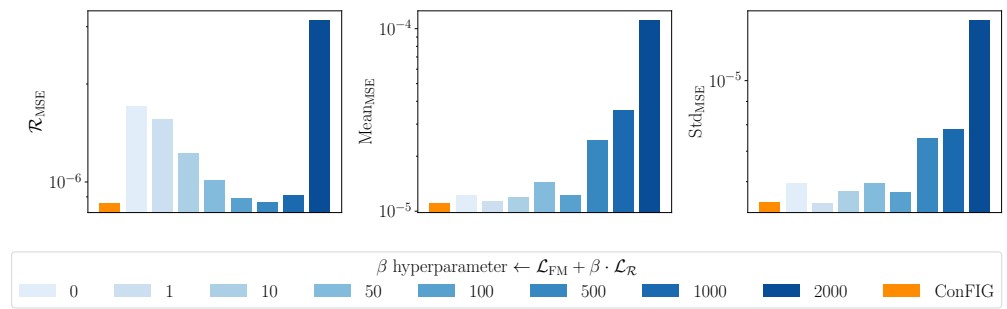

Figure 19: Comparison of MSE for physical residual, predicted mean, and standard deviation in the dynamic stall case, using ConFIG and fixed $\beta$ hyperparameters for loss weighting. The optimal configuration minimizes the error across all three metrics simultaneously.

*c.* We introduce two modifications compared to the original DiT implementation. First, we incorporate linear attention (Wang et al., 2020) alongside the standard quadratic one. Second, we replace the original label embedder with a two-layer module: a linear layer followed by a SiLU activation function, and a second linear layer that produces the final conditioning embedding.

All the hyperparameters are summarized in Table 9.

Table 9: Architecture and training hyperparameters for proposed test cases.

|  | **Darcy flow** | **Kolmogorov flow** | **Dynamic stall** |
|---|---|---|---|
| Training iterations | 1248000 | 512000 | 2048000 |
| Learning rate | $3 \cdot 10^{-5}$ | $1 \cdot 10^{-4}$ | $1 \cdot 10^{-4}$ |
| Batch size | 64 | 64 | 64 |
| Conditioning size | - | 1 | 4 |
| Output size | 2 | 2 | 6 |
| Patch size | 4 | 4 | 4 |
| Hidden size | 256 | 256 | 128 |
| DiT depth | 8 | 8 | 4 |
| Attention heads | 8 | 8 | 4 |
| Attention type | Quadratic | Linear | Linear |
| Parameters (M) | 9.75 | 9.81 | 1.29 |
| Gflops | 2.72 | 13.00 | 1.68 |

## H PERFORMANCE EVALUATION

Table 10 reports the inference wall-clock times for generating 32, 64, and 128 dynamic stall samples using 8 CPU cores of an Intel Xeon Platinum and an NVIDIA A100 64GB GPU. The same CPU was used to run the numerical simulations, ensuring a more consistent comparison of computational performance. On average, a single numerical simulation requires 76 minutes to complete. In contrast, the proposed generative model can produce 128 samples in just 2 and 4 minutes when using 10 and 20 FM steps on the CPU, respectively. When executed on a modern GPU, these inference times drop dramatically to 0.2 and 0.4 seconds, respectively. This substantial speedup highlights the model's potential as a fast and reliable surrogate for dynamic stall prediction in helicopter and wind turbine applications.

## I SAMPLER IMPLEMENTATION AND ADDITIONAL DETAILS

Algorithm 2 outlines the proposed sampling procedure, which is implemented using the explicit Euler integration scheme with $n$ equispaced time steps. The process begins by initializing $x_t$, which holds the sample values at time $t = 0$, with Gaussian noise. During the integration loop, each time

Table 10: Wall-clock time in seconds to generate $n$ samples using 8 cores of an Intel Xeon Platinum and an NVIDIA A100 64GB for dynamic stall case. Batch size is set equal to the number of samples. One simulation with the same cores takes on average 76 minutes ($4.56 \cdot 10^3$ s).

| FM | Number of samples - GPU | | | Number of samples - CPU | | |
|---|---|---|---|---|---|---|
| steps | 32 | 64 | 128 | 32 | 64 | 128 |
| 1 | $3.23 \cdot 10^{-3}$ | $3.38 \cdot 10^{-3}$ | $3.80 \cdot 10^{-3}$ | $2.27 \cdot 10^{0}$ | $6.16 \cdot 10^{0}$ | $1.20 \cdot 10^{1}$ |
| 2 | $9.76 \cdot 10^{-3}$ | $1.50 \cdot 10^{-2}$ | $2.53 \cdot 10^{-2}$ | $4.48 \cdot 10^{0}$ | $1.11 \cdot 10^{1}$ | $2.40 \cdot 10^{1}$ |
| 5 | $2.90 \cdot 10^{-2}$ | $4.95 \cdot 10^{-2}$ | $8.99 \cdot 10^{-2}$ | $1.17 \cdot 10^{1}$ | $2.96 \cdot 10^{1}$ | $5.99 \cdot 10^{1}$ |
| 10 | $6.15 \cdot 10^{-2}$ | $1.07 \cdot 10^{-1}$ | $1.98 \cdot 10^{-1}$ | $2.25 \cdot 10^{1}$ | $5.09 \cdot 10^{1}$ | $1.19 \cdot 10^{2}$ |
| 20 | $1.26 \cdot 10^{-1}$ | $2.23 \cdot 10^{-1}$ | $4.13 \cdot 10^{-1}$ | $4.68 \cdot 10^{1}$ | $1.15 \cdot 10^{2}$ | $2.40 \cdot 10^{2}$ |
| 50 | $3.21 \cdot 10^{-1}$ | $5.70 \cdot 10^{-1}$ | $1.06 \cdot 10^{0}$ | $1.12 \cdot 10^{2}$ | $3.08 \cdot 10^{2}$ | $5.99 \cdot 10^{2}$ |

step can be computed using either the standard deterministic FM sampler or the proposed stochastic variant. The choice between the two is governed by a user-defined boolean control parameter and is restricted to the initial segment of the trajectory, up to a threshold time $t^*$. In our experiments, we set $t^* = 0.2$, introducing additional stochasticity during the early phase of sampling while preserving high sample quality in later stages. In the stochastic sampler, the velocity $u_t^\theta$ is used to generate the final sample in a single forward step, followed by a backward update to time $t + dt$ using a new Gaussian noise sample.

---

**Algorithm 2** Deterministic and Stochastic Samplers

$dt \leftarrow 1/n$          $\triangleright$ $n$ is the number of integration steps
$x_t \leftarrow x_0 = \mathcal{N}(0, 1)$
**for** $i = 0$, $i < n$ **do**
    **if** $t < t^*$ **and** use stochastic sampler **then**
        $x_t \leftarrow x_t + (1 - t) \cdot u_t^\theta$          $\triangleright$ Integrate to $t = 1$
        $t \leftarrow t + dt$
        $x_t \leftarrow (1 - t) \cdot \mathcal{N}(0, 1) + t \cdot x_t$      $\triangleright$ Return to $t + dt$ using new generated normal noise
    **else**
        $x_t \leftarrow x_t + dt \cdot u_t^\theta$      $\triangleright$ Standard deterministic sampler, stochasticity is embedded in $x_0$
        $t \leftarrow t + dt$
    **end if**
**end for**

---

Our method introduces some bias during training, affecting the pure deterministic sampler ($t^* = 0.0$) and resulting in higher distributional error. However, for the stochastic sampler, the observed decrease in residual error and increase in distributional accuracy is also present for vanilla FM, indicating that this effect is not unique to our approach. The difference in distributional error in the deterministic sampler primarily arises due to conflicting gradients during training: when physical constraints and distributional objectives are jointly optimized, their gradients can oppose each other, leading to suboptimal updates for one or both objectives.

Table 11: Performance of stochastic sampler for Darcy flow problem over 1024 samples using 20 FM steps using FM-OT. Noise resampling is enabled for $t < t^*$, details in Algorithm 2. $t^* = 0.0$ coincides with deterministic sampler.

| Metric | $t^*$ | | | | | |
|---|---|---|---|---|---|---|
| | 0.0 | 0.2 | 0.4 | 0.6 | 0.8 | 1.0 |
| RE | 4.159 | 3.530 | 3.010 | 2.581 | 2.018 | 1.174 |
| WD $\cdot 10^2$ | 0.059 | 0.102 | 0.123 | 0.255 | 0.403 | 0.403 |
| JS $\cdot 10^1$ | 0.131 | 0.197 | 0.246 | 0.324 | 0.414 | 0.436 |

Table 12: Performance of stochastic sampler for Darcy flow problem over 1024 samples using 20 FM steps using FM-OT+ConFIG with residual loss. Noise resampling is enabled for $t < t^*$, details in Algorithm 2. $t^* = 0.0$ coincides with deterministic sampler.

| Metric | $t^*$ | | | | | |
|---|---|---|---|---|---|---|
| | 0.0 | 0.2 | 0.4 | 0.6 | 0.8 | 1.0 |
| RE | 1.331 | 1.448 | 1.317 | 1.152 | 1.002 | 0.765 |
| WD $\cdot 10^2$ | 0.391 | 0.096 | 0.139 | 0.254 | 0.383 | 0.361 |
| JS $\cdot 10^1$ | 0.293 | 0.193 | 0.266 | 0.320 | 0.395 | 0.404 |

To provide a complete picture, we report results for vanilla FM (Table 11) and for FM with the addition of a physical loss but without unrolling (Table 12), complementing the unrolling results presented in Table 3. The stochastic sampler helps prevent collapse to "unique" solutions, a phenomenon typical of PINN setups where deterministic optimization can lead to mode collapse and reduced sample diversity. By leveraging stochastic sampling, our method maintains higher distributional fidelity.

## J  PBFM VARIANTS

We performed two ablations on the dynamic stall benchmark to assess whether decomposing the physics objective or modifying gradient flow could yield further gains. The evaluated variants are:

- PBFM 3 losses: the physics objective is split into two separate residual losses, $\mathcal{R}_{ig}$ and $\mathcal{R}_\tau$, trained alongside the FM loss.
- PBFM last step: same as the base PBFM but the gradient is computed only at the last step of the unrolled predictions when computing the residual loss to reduce backward memory and computational time.

Table 13: Comparison of wall-clock time in seconds for one training iteration and memory usage in GB on an NVIDIA A100 64GB GPU for the proposed approaches. Batch size is 64 for all cases. Inference time is unchanged.

| Method | | 1 step | 2 steps | 3 steps | 4 steps |
|---|---|---|---|---|---|
| PBFM | [s] | $8.14 \cdot 10^{-2}$ | $1.18 \cdot 10^{-1}$ | $1.55 \cdot 10^{-1}$ | $1.90 \cdot 10^{-1}$ |
| | [GB] | 4.41 | 7.18 | 9.90 | 12.58 |
| PBFM | [s] | $1.08 \cdot 10^{-1}$ | $1.68 \cdot 10^{-1}$ | $2.29 \cdot 10^{-1}$ | $2.84 \cdot 10^{-1}$ |
| 3 losses | [GB] | 4.41 | 7.20 | 9.91 | 12.59 |
| PBFM | [s] | $8.35 \cdot 10^{-2}$ | $1.09 \cdot 10^{-1}$ | $1.23 \cdot 10^{-1}$ | $1.34 \cdot 10^{-1}$ |
| last step | [GB] | 4.41 | 8.48 | 8.48 | 8.48 |

Table 14: Generative performance metrics for dynamic stall problem using 20 FM steps. RE: physical Residual MSE, WD: Wassserstein Distance, JS: Jensen-Shannon divergence, MMSE: MSE of the mean fields, SMSE: MSE of the standard deviation fields.

| Dataset | Metric | PBFM | PBFM 3 losses | PBFM last step |
|---|---|---|---|---|
| | RE $\cdot 10^6$ | 0.339 | 0.315 | 0.392 |
| | WD $\cdot 10^4$ | 1.814 | 2.077 | 3.038 |
| Dynamic | JS $\cdot 10^2$ | 0.680 | 0.653 | 0.722 |
| Stall | MMSE $\cdot 10^5$ | 1.490 | 1.696 | 1.814 |
| | SMSE $\cdot 10^5$ | 0.874 | 0.842 | 0.809 |

Key findings are summarized in Tables 13 and 14. Splitting the physics terms into separate losses does not bring clear overall improvements. The three-loss variant attains a slightly lower residual

MSE in our runs, 0.315 vs. 0.339, while the Wasserstein distance increases from 1.814 to 2.077. Moreover, the three-loss setup increases per-iteration training time by roughly 30-40%, due to additional gradient computations; peak memory usage increases only marginally. Aggregating the physics constraints into a single residual loss and resolving conflicts via ConFIG provides the best trade-off between accuracy and training cost for the dynamic stall case.

The last step variant reduces the backward computation costs while the impacts on final accuracy are limited. We also measured gradient alignment between the two physics terms and found them largely aligned (average cosine similarity $\approx 0.76$), which explains why aggregating the physics terms into a single loss is both effective and more efficient in practice.

