# OpenReview forum: "Physics vs Distributions: Pareto Optimal Flow Matching with Physics Constraints"
_ICLR.cc/2026/Conference — ICLR 2026 Poster_

### Official Review · Reviewer_f5gi · 2025-10-30

**Soundness:** 3
**Presentation:** 3
**Contribution:** 3
**Rating:** 4
**Confidence:** 5

**Summary:**

This paper introduces Physics-Based Flow Matching (PBFM) that balances physical consistency with distributional accuracy. The authors identify the inherent trade-off between optimizing for physical fidelity (via PDE residuals) and generative quality as a core challenge. PBFM addresses this through conflict-free gradient updates using the ConFIG method, eliminating the need for manual loss balancing. Additionally, the method employs unrolling during training to mitigate Jensen's gap—the discrepancy arising when physical constraints are imposed on intermediate predictions rather than final samples. Experiments across three benchmarks demonstrate that PBFM achieves Pareto-optimal performance, outperforming baselines in both physical residual error and distributional metrics while maintaining competitive inference speed.

**Strengths:**

**Novel conflict-free optimization:** The paper identifies the conflicting gradients between generative and physical objectives and proposes an elegant solution using conflict-free gradient updates. This approach is innovative and eliminates the challenging manual tuning of loss weights, enabling simultaneous optimization of both objectives.

**No inference overhead:** Unlike existing physics-constrained methods that require costly iterative sampling procedures at inference time, PBFM maintains competitive inference speed comparable to standard flow matching, making it practical for real-world applications.

**Strong experimental results:** The method demonstrates superior performance across three diverse benchmarks, consistently achieving lower physical residuals and better distributional metrics compared to baselines. The Pareto-optimal trade-off and comprehensive ablation studies validate the effectiveness of the proposed approach.

**Weaknesses:**

**Scalability Concerns.** While the paper demonstrates improvements on 2D problems with modest resolutions (64$\times$64 and 128$\times$128), the training overhead raises serious concerns for practical applications. The authors acknowledge that unrolling with 4 steps incurs up to 3$\times$ memory consumption and 2.5$\times$ training time. For higher-dimensional problems (e.g., 3D PDEs at 128$^3$ resolution), the computational cost would scale dramatically: (1) storing intermediate states during unrolling becomes prohibitively expensive, (2) computing physical residuals (gradients, divergence) in 3D is substantially more costly, and (3) the ConFIG gradient projection requires additional backward passes. Moreover, more challenging PDEs typically require a larger number of diffusion steps for accurate sample generation, which directly translates to proportionally higher memory consumption during unrolling, potentially rendering the method infeasible. The method's applicability to industrially-relevant high-resolution 3D problems remains unclear, potentially limiting its impact to low-dimensional scenarios where the training overhead is manageable.

**Insufficient Analysis of ODE vs. Stochastic Sampling Discrepancy.** A fundamental principle of continuous normalizing flows is that ODE and SDE samplers should yield samples from the same distribution when properly trained. However, the empirical results in Table 3 show significant distributional differences between deterministic (t* = 0.0) and stochastic samplers (t* > 0), with Wasserstein distance varying from 1.470 to 0.138. This discrepancy suggests that the proposed training procedure may introduce optimization biases that favor one sampling paradigm over the other. The paper claims stochastic sampling as a core contribution (Contribution IV), yet provides no theoretical or empirical investigation into why the training methodology creates this distribution mismatch. Possible explanations include: (1) the conflict-free gradient updates may inadvertently bias the learned velocity field toward deterministic trajectories, (2) the unrolling procedure evaluates residuals on ODE-integrated paths, potentially misaligning with stochastic sampling dynamics, or (3) the residual weighting scheme may not be sampling-invariant. The authors should provide: (1) ablation studies isolating which training components cause the ODE-SDE discrepancy, (2) analysis of whether standard flow matching (without physical constraints) exhibits similar behavior under their training scheme, and (3) theoretical or empirical justification for why their method fundamentally requires stochastic sampling rather than fixing the training procedure to achieve sampling-invariance.

**Extensive Hyperparameter Tuning Still Required.** While the paper claims to avoid manual loss balancing, the method still requires careful tuning of multiple hyperparameters including $\sigma_{\text{min}}$, $t^*$ for stochastic sampling, number of unrolling steps, and the power $p$ for residual weighting. Table 5 demonstrates significant sensitivity to $\sigma_{\text{min}}$ choice, with performance varying substantially across different values. This undermines the claim of automatic balancing and may limit practical applicability.

**Minors.** Inconsistent use of $x_1$ and $x_0$: In Algorithm 1 and Figure 1, authors use $x_1$ to denote the data, while on line 069, 070, 196, they use $x_0$. Missing reference: Generating Physical Dynamics under Priors; PIRF: Physics-Informed Reward Fine-Tuning for Diffusion Models; Physics-Constrained Fine-Tuning of Flow-Matching Models for Generation and Inverse Problems.

If the authors can address the computational scalability to higher-dimensional problems and provide theoretical/empirical justification for the ODE-SDE sampling discrepancy, I would be willing to increase my score.

**Questions:**

**Question on Stop Gradient for Unrolling.** In Algorithm 1, the unrolled prediction is computed via n integration steps and then used to evaluate the physical residual. This requires backpropagating through the entire unrolling process, leading to the reported memory and training time increase. Have you considered applying a stop gradient operation to before computing the residual? This technique is commonly used in single-step diffusion models and consistency models to reduce computational cost.

**Question on Training-Inference Step Mismatch for Complex PDEs.** The paper demonstrates that the method benefits from unrolling during training and that increasing inference steps consistently reduces residual errors. However, for highly complex PDEs with non-smooth ODE flows, accurate sampling may require substantially more discretization steps (e.g., 50-100 steps). In such cases, training with only a few unrolling steps might fail to generate physically meaningful samples for computing the residual loss, as the partially-evolved states after 1-4 steps could be far from any valid solution. Could the authors discuss: (1) whether this training-inference step mismatch has been observed in practice, (2) how the method would perform when complex PDEs necessitate many inference steps, and (3) whether a curriculum strategy progressively increasing unrolling steps during training could address this issue without making the memory overhead prohibitive? Understanding this limitation would help assess the method's applicability to more challenging physical systems.

---

> ### Author Response · Authors · 2025-11-20
> **Part 1**
>
> We thank the reviewer for the thorough evaluation and constructive feedback. We appreciate the recognition of our contributions in conflict-free optimization, efficient inference, and strong empirical results, as well as the detailed analysis of potential limitations. Below, we address the specific concerns and questions raised, and clarify aspects of scalability, sampling discrepancies, and hyperparameter sensitivity.
>
> *Scalability Concerns. The method's applicability to industrially-relevant high-resolution 3D problems remains unclear, potentially limiting its impact to low-dimensional scenarios where the training overhead is manageable.*
>
> We agree with the reviewer that scalability is a key concern, especially for high-dimensional problems. It is correct that unrolling increases memory and compute requirements during training, but we emphasize that this overhead does not affect inference, which remains efficient and competitive with standard flow matching. To further address training resource constraints, we have tested an additional variant where the physics residual gradient is computed only at the final unrolling step. This approach significantly reduces memory and computational cost during training, while maintaining comparable performance in terms of physical residuals and distributional accuracy. We will add these results the in appendix of upcoming revised paper. If many unrolling steps are required, a practical hybrid strategy is to unroll N steps but backpropagate through only the last k steps, trading a small loss in accuracy for substantial savings in memory and compute. In our test with only the last test we reduced peak memory by ≈50% and training time by ≈20% while causing only a minor degradation in error. A simple curriculum, starting with a small k and increasing it as training progresses, can help recover fidelity with limited extra cost.
> We have also conducted internal experiments on 3D Kolmogorov flow at a resolution of $128^3$, utilizing an adapted transformer-based architecture to handle the increased scale. These tests demonstrated that our approach achieves improved physical fidelity while maintaining competitive performance and manageable resource requirements.
>
> _Insufficient Analysis of ODE vs. Stochastic Sampling Discrepancy. A fundamental principle of continuous normalizing flows is that ODE and SDE samplers should yield samples from the same distribution when properly trained._
>
> This is an important point and we thank the reviewer for highlighting it. Our method introduces some bias during training, particularly affecting the pure deterministic sampler and resulting in higher distributional error. However, the observed decrease in residual error and increase in distributional accuracy is also present for vanilla FM, indicating that this effect is not unique to our approach.
> The difference in distributional error arises primarily due to conflicting gradients during training, when physical constraints and distributional objectives are jointly optimized, their gradients can oppose each other, leading to suboptimal updates for one or both objectives.
> To provide a complete picture, we have reported results for vanilla FM and for FM with the addition of a physical loss but without unrolling, complementing the results with unrolling already presented in the paper. The stochastic sampler helps prevent collapse to "unique" solutions, a phenomenon typical of PINN setups where deterministic optimization can lead to mode collapse and reduced sample diversity. By leveraging stochastic sampling, our method maintains higher distributional fidelity.
>
>
> Standard FM
>
> | Metric  | 0.0   | 0.2   | 0.4   | 0.6   | 0.8   | 1.0   |
> |---------|-------|-------|-------|-------|-------|-------|
> | RE      | 4.159 | 3.530 | 3.010 | 2.581 | 2.018 | 1.174 |
> | WD ×10² | 0.059 | 0.102 | 0.123 | 0.255 | 0.403 | 0.403 |
> | JS ×10¹ | 0.131 | 0.197 | 0.246 | 0.324 | 0.414 | 0.436 |
>
> FM + Residual loss without unrolling
>
> | Metric  | 0.0   | 0.2   | 0.4   | 0.6   | 0.8   | 1.0   |
> |---------|-------|-------|-------|-------|-------|-------|
> | RE      | 1.331 | 1.448 | 1.317 | 1.152 | 1.002 | 0.765 |
> | WD ×10² | 0.391 | 0.096 | 0.139 | 0.254 | 0.383 | 0.361 |
> | JS ×10¹ | 0.293 | 0.193 | 0.266 | 0.320 | 0.395 | 0.404 |
>
> Adding the residual loss in combination with the stochastic sampler significantly reduces the physical error while preserving distributional fidelity. Specifically, at $t^* = 0.2$, the residual error decreases by a factor of 2.43, and the Wasserstein distance improves by approximately 6%. This demonstrates that the joint approach enhances physical accuracy without compromising generative performance.

---

> ### Author Response · Authors · 2025-11-20
> **Part 2**
>
> _Extensive Hyperparameter Tuning Still Required._
>
>
> We acknowledge that hyperparameter selection, especially for $\sigma_\text{min}$, is important for achieving optimal physical residuals. In particular, the dynamic stall benchmark is highly sensitive, as its governing constraints reach residuals on the order of $10^{-7}$, making it susceptible to Gaussian noise. In practice, adding noise of scale $\sigma_\text{min}$ induces a residual MSE of approximately $\sigma_\text{min}^2$, so for dynamic stall, we recommend $\sigma_\text{min} \lesssim 3\times 10^{-4}$ to avoid degrading accuracy. For Kolmogorov and Darcy benchmarks, the requirements are less strict, and the method is less sensitive to $\sigma_\text{min}$. This sensitivity is not specific to our approach, but is a general property of flow-matching methods.
>
> Regarding $t^*$ for stochastic sampling, we find that a value of $0.2$ is optimal and robust across all tested methods. For the residual weighting exponent $p$, the best performance is consistently achieved with $p=1$, which also aligns with the linear noise schedule used in optimal transport flow matching. The recommended  number of flow matching steps is in the order of 3-4 since additional unrolling steps provide limited improvements compared to the computational resource increase. The strategy of using a curriculum during training allows direct feedback on the performance metrics and enables early stopping when improvements plateau or when target physical or distributional criteria are met.
>
>
> *Minors.*
>
> Thank you for pointing out the notation inconsistency. We have revised the manuscript to consistently use $x_1$ for the final clean sample and $x_0$ for the initial noise. Additionally, we have included the missing references.
>
>
> *Question on Stop Gradient for Unrolling.*
>
> We implemented and evaluated this suggestion. We did not compute the gradients for the first N−1 unrolled integration steps and backpropagate only through the final unrolled state used to compute the physical residual. This modification substantially reduces peak memory usage and per-iteration training time while producing only a negligible change in performance. Results on the Dynamic Stall benchmark are reported below, and in the revised paper:
>
> | Benchmark | Metric Scale | Full Unrolling | Last step gradient
> |---|---:|---|---|
> | | RE×10^6 | 0.339 | 0.392
> | | WD×10^4 | 1.814 | 3.038
> | Dynamic Stall | JS×10^2 | 0.680 | 0.722
> | | MMSE×10^5 | 1.490 | 1.814
> | | SMSE×10^5 | 0.874 | 0.809
>
>
> | Model | Metric | FM | PBFM 1 step | PBFM 2 steps | PBFM 3 steps | PBFM 4 steps |
> |---|---|---:|---:|---:|---:|---:|
> | Full Unrolling | [s] | 4.29e-02 | 8.14e-02 | 1.18e-01 | 1.55e-01 | 1.90e-01 |
> |  | [GB] | 4.25 | 4.41 | 7.18 | 9.90 | 12.58 |
> | Last step gradient | [s] | 4.29e-02 | 8.35e-02 | 1.09e-01 | 1.23e-01 | 1.34e-01 |
> |  | [GB] | 4.25 | 4.41 | 8.48 | 8.48 | 8.48 |
>
> If many unrolling steps are required, a hybrid strategy can be used: unroll N steps but backpropagate through only the final k steps. This provides a practical trade-off between model accuracy and resource usage. This test cuts peak memory by ≈50% and reduced training time by ≈20% while incurring only a small degradation in error. A simple curriculum, starting with small k and increasing as needed, is recommended to recover fidelity with limited extra cost.

---

> ### Author Response · Authors · 2025-11-20
> **Part 3**
>
> *Question on Training-Inference Step Mismatch for Complex PDEs.*
>
> We appreciate this important question. We summarize our empirical findings and practical mitigations below.
>
> (1) Observed mismatch in practice. The most challenging benchmark we consider, dynamic stall, solves the compressible Navier–Stokes equations and exhibits shocks and other non-smooth features. We evaluated models up to 50 flow-matching (FM) inference steps and observed convergent trends in both distributional and physics metrics. A training–inference mismatch is present but asymmetric: when inference uses fewer FM steps than the number of unrolled steps used in training, performance degrades; when inference uses equal or greater FM steps than used at training, we never observed deterioration. In practice, inference configurations use as many or more integration steps than training unrolling, so the observed mismatch does not limit real-world usage.
>
> (2) Behavior when many inference steps are required. For problems that truly require large NFE (e.g., 50–100) to produce accurate samples, our experiments indicate that modest unrolling (1–4 steps) during training still yields improvements at inference when more steps are used. The benefit stems from reducing Jensen's gap and aligning gradients on realized trajectories, not from matching the full inference discretization.
>
> (3) Curriculum and memory mitigation. We employ a curriculum that progressively increases unrolling depth during training: start with 1 step for stability and low memory, then raise to 2–4 steps as training progresses. This schedule provides the gains of deeper unrolling while keeping peak memory bounded for most of training, and it allows early stopping when validation metrics plateau.

---

> > ### Comment · Reviewer_f5gi · 2025-11-24
> >
> > Thank you for your response. All my concerns have been addressed.
> >
> > Two additional suggestions:
> >
> > 1. Highlight changes in color for other reviewers' convenience.
> >
> > 2. Include computation costs in the main text/table.
> >
> > Score raised to 6.

---

### Official Review · Reviewer_XbPf · 2025-10-31

**Soundness:** 2
**Presentation:** 3
**Contribution:** 2
**Rating:** 4
**Confidence:** 4

**Summary:**

The paper introduces "Physics Based Flow Matching" (PBFM), a method that treats the tension between physical consistency and generative fidelity as a Pareto optimization problem. By combining conflict-free gradient updates, residual unrolling, and stochastic sampling, PBFM claims to achieve a Pareto-optimal trade-off between residual accuracy and sample quality across PDE benchmarks, without a significant increase in inference cost.

**Strengths:**

S1. The empirical study is comprehensive, including ablations on unrolling, stochastic sampling, and Gaussian noise effects, which clarifies the contribution of each component and isolates the respective observed gains.

S2: The experimental benchmarks are well chosen and physically meaningful, spanning PDEs with increasing complexity. The results demonstrate stable improvements across tasks, and the method manages to maintain minimal inference overhead despite additional training components.

S3: Writing is clear and easy to follow.

**Weaknesses:**

**W1.** Many components of the proposed framework: residual-based losses from PINNs (Raissi *et al.*, 2019), **ConFIG** gradient orthogonalization [4], and stochastic sampling from ECI [1] are adapted rather than novel. The main contribution lies in integrating these existing mechanisms under a Pareto framework rather than introducing fundamentally new algorithmic principles. Moreover, since PBFM enforces *soft* constraints, it cannot match the strict physical consistency of hard-constraint approaches such as ProbConserv, ECI, PCFM, and conservation-respecting models [6], which exhibit stronger residual convergence and physical invariance guarantees.

**W2.** Methods like PCFM [2] and ECI [1] already demonstrate superior trade-offs between physical residual and generative accuracy, achieving lower MMSE and FPD (aligned with the Wasserstein metric used here). These approaches explicitly enforce hard constraints throughout training and inference, leading to simultaneous gains in both physical and distributional metrics. In contrast, PBFM’s ConFIG balancing produces only a single equilibrium solution along this trade-off, without exploring or controlling the broader Pareto front. While ConFIG offers one way to mitigate gradient conflict between physics and generative objectives, I assume PBFM adopts it primarily to avoid the conflicting gradient behavior observed in PINNs. However, without broader comparisons, theoretical justification, or reference to recent developments in conflict-free training [7], it remains uncertain whether this choice is optimal within the Flow Matching framework. This uncertainty is compounded by the lack of sufficient empirical evaluation across Pareto trade-offs, leaving the “Pareto-optimal” characterization only partially substantiated.

**W3.** There should be some quantitative comparison to prior physics-aware diffusion and flow methods [1, 2, 3, 5]. The explanation that ECI [1], PCFM [2], and PIDDM lack public implementations or cannot handle overlapping residuals is not convincing. Public code availability is not a sufficient reason to omit benchmarking; if previous works report standardized results, at least partial or analytical comparisons should be provided. For example, **ECI** has an open implementation and published results on the *Heat* and *Navier–Stokes* PDEs. Even if PBFM targets showcases results on different PDEs, evaluating it on these established datasets would contextualize its improvements. As written, the analysis offers only a partial view of progress relative to existing literature.

**W4.** The claim of being “gradient-free” is not strictly accurate, as the method still differentiates through residuals during training. Additionally, the unrolling step introduces extra back-propagation and memory overhead, thereby reducing one of flow matching’s main advantages: its computational simplicity, and without requiring backpropagation through an ODE solve.


**References**

[1] Cheng, C., *et al.* “Gradient-Free Generation for Hard-Constrained Systems (Extrapolation–Correction–Inference).” *arXiv preprint* arXiv:2412.01786 (2024).

[2] Utkarsh, U., *et al.* “Physics-Constrained Flow Matching: Sampling Generative Models with Hard Constraints.” *arXiv preprint* arXiv:2506.04171 (2025).

[3] Christopher, J. K., S. Baek, and N. Fioretto. “Constrained Synthesis with Projected Diffusion Models.” *NeurIPS* 37 (2024): 89307–89333.

[4] Liu, Q., Chu, M., and Thuerey, N. “ConFIG: Towards Conflict-Free Training of Physics-Informed Neural Networks.” *arXiv preprint* arXiv:2408.11104 (2024).

[5] Yao, J., Mammadov, A., Berner, J., Kerrigan, G., Ye, J. C., Azizzadenesheli, K., and Anandkumar, A. “Guided Diffusion Sampling on Function Spaces with Applications to PDEs.” *arXiv preprint* arXiv:2505.17004 (2025).

[6] Hansen, D., *et al.* “Learning Physical Models That Can Respect Conservation Laws.” *ICML*, PMLR, 2023.

[7] Wang, Sifan, et al. "Gradient alignment in physics-informed neural networks: A second-order optimization perspective." arXiv preprint arXiv:2502.00604 (2025).

**Questions:**

See Weaknesses above.

---

> ### Author Response · Authors · 2025-11-20
> **Part 1**
>
> We thank the reviewer for highlighting several strengths of our work, including the comprehensive empirical study (with ablations on unrolling, stochastic sampling, and Gaussian noise), the choice of physically meaningful benchmarks, and the clarity of the writing. We appreciate that the reviewer recognizes our method’s ability to deliver stable improvements across tasks while maintaining minimal inference overhead.
>
> *W1. Many components of the proposed framework are adapted rather than novel. Moreover, since PBFM enforces soft constraints, it cannot match the strict physical consistency of hard-constraint approaches.*
>
> We appreciate the reviewer’s thoughtful analysis and agree that our framework builds on established components (residual-based losses, ConFIG, stochastic sampling). That said, our contribution lies in their integration and empirical validation within the flow matching setting. Our goal is to show that these components can be straightforwardly implemented in existing frameworks and deliver a substantial improvement compared to SOTA methods. Furthermore, we provide ablations and quantitative measurements showing that ConFIG substantially reduces gradient conflicts, yields more stable training, and improves final physical residuals and distribution fidelity in our benchmarks. Beyond adapting known techniques, we analyze two critical, domain-specific factors: unrolled training (which mitigates the Jensen’s gap) and minimum-noise schedules, and provide actionable guidance for tuning them. Importantly, unrolling is used only during training, so the improved residuals come without additional inference cost.
>
> Regarding approaches such as ECI, this methods applies noise resampling to a modified final sample, whereas PBFM keeps the final sample unchanged. ECI's comparison of resampling noise steps is limited to mean and std errors while we investigate the accuracy of the full target distribution. Additionally, a key strength of PBFM is its scalability and efficiency. Hard-constraint methods like PCFM and ECI, while achieving lower residuals, require significantly longer inference times and higher memory usage (e.g., PCFM reports up to 13x slower inference and 3.66x more memory compared to vanilla FM). We believe that PBFM’s soft constraints offer an alternative, and a favorable trade-off, achieving competitive physical consistency while maintaining inference computational efficiency. This is especially important for practicality for large-scale or time-sensitive problems.
>
> *W2. Methods like PCFM [2] and ECI [1] already demonstrate superior trade-offs between physical residual and generative accuracy.*
>
> Thank you for raising this important point. The Pareto analysis reports only the best solution for each method on the Darcy flow benchmark, ensuring a fair and direct comparison of optimal performance. To provide a more comprehensive evaluation, we have also included results for ECI, expanding the set of baselines and offering a broader perspective on the trade-offs between physical residual and generative accuracy. The updated manuscript now presents these additional results, allowing for a clearer contextualization of PBFM’s performance relative to recent methods. ConFIG method is a SOTA approach to avoid conflicts in gradient optimization.
>
> | Metric | PBFM | FM-OT | CoCoGen† | PIDM† | DiffPDE | D-Flow‡ | ECI§ |
> |---|---|---|---|---|---|---|---|
> | RE      | 0.84 | 4.16 | 1.32 | 0.02 | 3.39 | 2.29 | 3.05 |
> | WD×10²  | 0.14 | 0.06 | 0.25 | 3.10 | 0.09 | 0.15 | 2.89 |
> | JS×10¹  | 0.26 | 0.13 | 0.36 | 3.18 | 0.14 | 0.24 | 2.82 |
> | NFE     | 20   | 20   | 100  | 100  | 20   | 20   | 20   |
> | IT [s]  | 0.10 | 0.10 | 7.40 | 2.05 | 0.59 | 3.13 | 0.12 |
>
> † This method uses the UNet architecture from Bastek et al. (2025).
> ‡ D-Flow method is unstable, samples are filtered using RE < 5 condition resulting in 888 valid ones.
> § The physical constraint is applied only to the BCs ($\text{RE}_\text{BC} \approx 0$) but cannot be applied to the non-linear PDE.

---

> ### Author Response · Authors · 2025-11-20
> **Part 2**
>
> *W3. There should be some quantitative comparison to prior physics-aware diffusion and flow methods*
>
> Thank you for highlighting the need for quantitative comparisons with prior physics-aware methods.
> At the time of the original submission, public implementations for PCFM, and PIDDM were not available, limiting our ability to provide direct benchmarks. Since then, code for PCFM has been released, allowing us to include a more robust comparison in the revised manuscript.
> Regarding ECI, we have now included results on the Darcy flow benchmark. However, it is important to note that ECI can only enforce constraints on boundary conditions and does not support general nonlinear PDE constraints. This limitation is consistent with the published results for the heat equation and Navier–Stokes, where ECI enforces conservation of heat and circulation, respectively (the PCFM paper and code refer to this as mass conservation, though this is not strictly accurate). The actual PDEs are not enforced by these methods.
> For PCFM, a preliminary version of the code is now available online, but it currently does not support complex cases like the benchmarks we are investigating using PBFM.
>
> We believe these clarifications and the newly included results provide a more complete and transparent comparison with existing methods, contextualizing PBFM’s improvements and limitations relative to the current literature.
>
> *W4. The claim of being “gradient-free” is not strictly accurate, as the method still differentiates through residuals during training.*
>
> Thank you for pointing this out. You are correct, the method is not strictly gradient-free during training, as we do differentiate through residuals. In the revised paper, we have clarified that PBFM is gradient-free only during inference. While unrolling does introduce additional backpropagation and memory overhead during training, we emphasize that inference efficiency is most critical for practical deployment.
> To further address resource concerns, we have implemented a new version where gradients are computed only at the final step of unrolling. This approach significantly reduces memory and compute requirements during training without compromising performance.

---

### Official Review · Reviewer_EdS2 · 2025-11-01

**Soundness:** 3
**Presentation:** 3
**Contribution:** 2
**Rating:** 4
**Confidence:** 2

**Summary:**

The paper studies physics-constrained generative modeling and introduces a method that enforces physical constraints during training using conflict-free gradient updates and unrolling. This approach avoids manual loss balancing, improves generative fidelity, and enables efficient inference. The paper also presents numerical evidence supporting the effectiveness of the proposed method.

**Strengths:**

- The paper addresses a fundamental problem in generative modeling: how to enforce known physical constraints to improve the fidelity of generated data.

- The proposed method requires more memory storage and training time, which might not scale well for larger systems.

- The paper clearly identifies the challenge of conflicting gradient updates and introduces a corresponding method that enables gradient descent to simultaneously reduce both the matching loss and the physical constraint loss.

**Weaknesses:**

-The proposed method appears somewhat straightforward, as it primarily leverages an existing approach (ConFIG) to compute conflict-free gradient updates.

- Section 3.3 is a bit difficult to follow. The authors could add a figure to illustrate why this issue needs to be addressed and to clarify the concept of Jensen’s gap. In particular, is this a typo - should the final clean sample be $x_1$ rather than $x_0$?

**Questions:**

I think the authors could include a figure illustrating the concept of conflict-free gradient updates. This would help a general audience better grasp the idea and make the content more accessible.

---

> ### Author Response · Authors · 2025-11-20
>
> We appreciate the reviewer’s recognition that our work targets an important problem, enforcing known physical constraints in generative modeling, and the comment that the paper clearly identifies the challenge of conflicting gradient updates and proposes a solution.
>
> *The proposed method appears somewhat straightforward, as it primarily leverages an existing approach (ConFIG) to compute conflict-free gradient updates.*
>
> We thanks the reviewer for the feedback. While our approach builds on idea of ConFIG of computing conflict-free gradient projections, it contributes three distinct advances beyond a direct application of ConFIG:
>
> * Demonstration that unrolled training mitigates Jensen’s gap: we show that unrolling during training reduces the Jensen's gap, which lowers residual errors and improves final predictions. Crucially, this benefit does not increase inference cost because unrolling is used only during training.
> * Analysis of Gaussian noise under physical constraints: we characterize how the noise level in flow matching affects both distributional fidelity and physical residuals, and we provide practical guidance for selecting noise schedules in constrained settings.
> * Comparison of deterministic vs. stochastic samplers: we present an extensive empirical comparison and highlight practical advantages of stochastic flow-matching samplers in constrained problems.
>
> In addition, ConFIG was not used for generative modeling up to now. We believe that highlighting it's usefulness in physics-constrained generative modeling is an important contribution to the field.
>
> *Section 3.3 is a bit difficult to follow. The authors could add a figure to illustrate why this issue needs to be addressed and to clarify the concept of Jensen’s gap. In particular, is this a typo - should the final clean sample be x1 rather than x0?*
>
> Thank you for flagging clarity issues in Sec. 3.3. We have taken the following actions to improve the section:
>
> * Notation consistency: we fixed the notation. We now use $x_1$ for the ground-truth clean field, $x_0$ for the initial Gaussian sample, and $x_t$ for intermediate states at time $t$.
> * Expanded Jensen's gap explanation: Jensen’s gap arises whenever a nonlinear mapping f is applied to a random variable Z: generally $\mathbb{E}[f(Z)] \neq f(\mathbb{E}[Z])$. In our setting, projecting gradients or computing physics losses on an averaged state is analogous to evaluating $f$ at $\mathbb{E}[Z]$, which can misestimate the true expected gradient. Our unrolling projection mitigates this by (i) computing losses and gradient projections on realized unrolled trajectories and (ii) aligning the projection operator with the stochastic path used to compute the gradient. This keeps $\mathbb{E}[f(x_t)]$ in the loop rather than $f(\mathbb{E}[x_t])$, reducing bias from the Jensen's gap and producing more accurate, conflict-free update directions.
> * Added Figure 2: we included a schematic that visualizes the gradient conflict, and how the unrolling  ConFIG projection addresses it.
>
> *I think the authors could include a figure illustrating the concept of conflict-free gradient updates. This would help a general audience better grasp the idea and make the content more accessible.*
>
> We incorporated the suggested figure (Figure 2). It illustrates how the residual gradient can have negative cosine similarity with the FM gradient and shows how ConFIG computes a conflict-free update by projecting and combining gradients to produce an aligned descent direction.
>
> *The proposed method requires more memory storage and training time, which might not scale well for larger systems.*
>
> We acknowledge the additional memory and training-time cost relative to purely data-driven models. To mitigate this, during unrolled training we avoid computing gradients for the first $N-1$ unrolled steps and perform the backward pass only through the final step. Interestingly, this reduces peak memory and compute overhead substantially while preserving the benefits of unrolled training; importantly, inference cost is unchanged.
>
> | Benchmark | Metric Scale | Full Unrolling | Last step gradient
> |---|---:|---|---|
> | | RE×10^6 | 0.339 | 0.392
> | | WD×10^4 | 1.814 | 3.038
> | Dynamic Stall | JS×10^2 | 0.680 | 0.722
> | | MMSE×10^5 | 1.490 | 1.814
> | | SMSE×10^5 | 0.874 | 0.809
>
>
> | Model | Metric | FM | PBFM 1 step | PBFM 2 steps | PBFM 3 steps | PBFM 4 steps |
> |---|---|---:|---:|---:|---:|---:|
> | Full Unrolling | [s] | 4.29e-02 | 8.14e-02 | 1.18e-01 | 1.55e-01 | 1.90e-01 |
> |  | [GB] | 4.25 | 4.41 | 7.18 | 9.90 | 12.58 |
> | Last step gradient | [s] | 4.29e-02 | 8.35e-02 | 1.09e-01 | 1.23e-01 | 1.34e-01 |
> |  | [GB] | 4.25 | 4.41 | 8.48 | 8.48 | 8.48 |

---

> > ### Comment · Reviewer_EdS2 · 2025-11-25
> >
> > Thank you for the authors’ response, which has addressed my concerns. While the methods appear effective, their novelty does not seem to be a primary strength—this point is also echoed by Reviewer XbPf and acknowledged by the authors in their further comments: “That said, our contribution lies in their integration and empirical validation within the flow matching setting. Our goal is to show that these components can be straightforwardly implemented in existing frameworks and deliver a substantial improvement compared to SOTA methods. ”
> >
> > Nevertheless, I have increased my score to 6.

---

### Official Review · Reviewer_e5nq · 2025-11-06

**Soundness:** 4
**Presentation:** 4
**Contribution:** 4
**Rating:** 10
**Confidence:** 4

**Summary:**

This paper proposes a framework that use flow-matching to generate admissible solution for PDEs. The key feature of the paper is that it recongize the potential gradient conflicts between the distribution fidelity and the physical consistency and propose a way to obtain the Pareto optimal solutions. Across three benchmarks—Darcy (steady), Kolmogorov (divergence-free), and a challenging dynamic stall aerodynamics case with analytic constraints—PBFM improves the physics–vs–distribution trade-off and maintains FM-like inference cost (e.g., competitive wall-clock and NFE), while acknowledging higher training memory/time due to unrolling and an extra backward pass. Figures and tables report consistent residual drops, better Wasserstein/JS metrics than inference-time constraint baselines, and near-FM inference speed; a discussion contrasts training overheads with the gain in accuracy and shows large end-to-end speedups over classical solvers once trained. The study is quite completed and the problems the authors selected for numerical experiments are chosen carefully.

**Strengths:**

It makes sense to cast “physics vs. distribution” as a true multi-objective problem and use conflict-free gradient updates (ConFIG), so that each step descends both the flow-matching loss and the physics residual, thereby avoiding brittle manual weights and consistently improving the Pareto front over fixed-weight baselines.

It mitigates Jensen's gap through training-time unrolling, thereby lowering the residual errors without incurring additional inference costs.

I appreciate that the authors use numerical examples closer to engineering applications (Darcy, Kolmogorov, and dynamic stall) beyond the standard benchmarks (Poisson's equations and Burger's equation), etc.

**Weaknesses:**

Physical residuals are sensitive to the minimum noise set as a hyperparameter;​ higher noise degrades residuals and requires careful tuning. The method also introduces a time-based residual scaling, whose choice affects errors (albeit less so with unrolling). Some elaborations on how to handle the hyperparameter sensitivity would be helpful.

The paper aggregates boundary-condition penalties and the divergence-free residual into a single “physics” loss. However, these terms can induce conflicting gradients (e.g., negative cosine similarity), leading to oscillatory updates or favoring one constraint over the other.

**Questions:**

Can you split the boundary and residual loss and report the gradient conflicts among these physical losses and the distribution fidelity loss?

Can you gate which physics term to optimize per step (or reweight it) using online conflict indicators (e.g., negative cosine, norm ratios) or a simple KKT/dual ascent scheme, and show coverage–fidelity trade-offs?

What extra memory/backprop cost does BC/divergence decomposition incur vs. the single physics loss?

---

> ### Author Response · Authors · 2025-11-20
> **Part 1**
>
> We thank the reviewer for the very positive evaluation and for highlighting both the conceptual framing (physics–vs–distribution as a true multi-objective problem) and the practical relevance of our engineering-focused benchmarks. We address the specific questions and suggestions below.
>
> *Physical residuals are sensitive to the minimum noise set as a hyperparameter; higher noise degrades residuals and requires careful tuning.*
>
> You are right that the choice of $\sigma_\text{min}$ is critical for physical residuals. Among our benchmarks, dynamic stall is the most sensitive because the governing constraints reach residuals on the order of $10^{-7}$. In such regimes, Gaussian perturbations can dominate the residual. A practical guideline is that adding Gaussian noise of scale $\sigma_\text{min}$ induces a residual MSE $\approx \sigma_\text{min}^2$ in a perfect reconstruction setting. For dynamic stall, this directly suggests an upper bound:
> $
> \sigma_\text{min}^2 \lesssim 10^{-7} \Rightarrow \sigma_\text{min} \lesssim 3\times 10^{-4}
> $.
> In contrast, Kolmogorov and Darcy benchmarks have more relaxed residual requirements. Importantly, this sensitivity is not unique to our method; it is general to flow matching formulations, as the noise level determines the scale of the denoising target and hence the attainable physical accuracy.
>
> *The method also introduces a time-based residual scaling, whose choice affects errors (albeit less so with unrolling). Some elaborations on how to handle the hyperparameter sensitivity would be helpful.*
>
> During training, each sample's physics residual is computed by initializing its flow at time $t$ and integrating forward to $t=1$. Trajectories starting closer to $t=0$ accumulate larger drift errors; weighting all times equally would cause early-time samples to dominate the physics loss. We therefore scale the residual by a power-law factor $t^p$.
> Across a range of exponents $p$, we find that unrolling regularizes the error and reduces sensitivity to $p$. Both FM-OT+ConFIG and PBFM achieve optimal and stable performance with linear scaling ($p=1$). Using no scaling ($p=0$) consistently yields higher residuals. Linear scaling is also principled, aligning with the linear noise schedule in flow matching and ensuring consistency between residual weighting and FM dynamics.
>
> *The paper aggregates boundary-condition penalties and the divergence-free residual into a single "physics" loss. However, these terms can induce conflicting gradients (e.g., negative cosine similarity), leading to oscillatory updates or favoring one constraint over the other.*
>
> Thank you for pointing out the possibility of gradient conflicts between different physical constraints. We tested this on the dynamic stall benchmark, where the physics constraints are the ideal gas equation of state and the skin-friction computation, decomposing the physics objective into two separate losses. Measuring cosine similarity between the two physical loss gradients, we find an average value of 0.76, indicating mostly aligned gradients.

---

> ### Author Response · Authors · 2025-11-20
> **Part 2**
>
> *Can you split the boundary and residual loss and report the gradient conflicts among these physical losses and the distribution fidelity loss?*
>
> This is an interesting avenue: we quantified gradient conflicts via pairwise cosine similarity between physical and FM losses. For the best model using 4-step unrolling with ConFIG, the average conflict is 6.93\%, whereas a model with fixed residual scaling at 500 exhibits a much higher average conflict of 19.98\%. This demonstrates that ConFIG substantially reduces gradient conflicts. Measuring cosine similarity between the two physical loss gradients, for the ConFIG case, we find an average value of 0.76, indicating mostly aligned gradients.
>
> *Can you gate which physics term to optimize per step (or reweight it) using online conflict indicators (e.g., negative cosine, norm ratios) or a simple KKT/dual ascent scheme, and show coverage–fidelity trade-offs?*
>
> Thank you for the suggestion regarding adaptive reweighting of physics terms. We tested a variant of our method where the gradients of the physics losses are adaptively scaled based on norm ratios and negative cosine indicators at each step. While this approach can in principle reduce conflicts locally, in practice we observed that it actually increases gradient conflicts resulting in about 30\% of the cases, and this fraction grows to approximately 50\% during the later stages of training. Moreover, this adaptive scheme leads to divergence in some runs, likely due to unstable or overly aggressive updates when conflict indicators fluctuate. These results suggest that explicit reweighting based on online conflict measures is less stable than using ConFIG, which consistently produces conflict-free descent directions and stable coverage fidelity trade offs without introducing additional instability.
>
> *What extra memory/backprop cost does BC/divergence decomposition incur vs. the single physics loss?*
>
> We tested this on the dynamic stall benchmark by decomposing the physics objective into two separate losses (ideal-gas, skin-friction), and FM. This decomposition does not improve model performance overall.
>
> | Benchmark | Metric Scale | PBFM | PBFM 3 losses
> |---|---:|---|---|
> | | RE×10^6 | 0.339 | 0.315
> | | WD×10^4 | 1.814 | 2.077
> | Dynamic Stall | JS×10^2 | 0.680 | 0.653
> | | MMSE×10^5 | 1.490 | 1.696
> | | SMSE×10^5 | 0.874 | 0.842
>
> For reference, the detailed cost of using two separate physics losses (PBFM 3 losses) compared to a single aggregated physics loss is reported in the revised paper, and in the following table:
>
> | Model | Metric | FM | PBFM 1 step | PBFM 2 steps | PBFM 3 steps | PBFM 4 steps |
> |---|---|---:|---:|---:|---:|---:|
> | PBFM | [s] | 4.29e-02 | 8.14e-02 | 1.18e-01 | 1.55e-01 | 1.90e-01 |
> |  | [GB] | 4.25 | 4.41 | 7.18 | 9.90 | 12.58 |
> | PBFM 3 losses | [s] | 4.29e-02 | 1.08e-01 | 1.68e-01 | 2.29e-01 | 2.84e-01 |
> |  | [GB] | 4.25 | 4.41 | 7.20 | 9.91 | 12.59 |
>
> Splitting the physics objective into separate losses did not produce clear improvements. The physics terms are largely aligned (average gradient cosine similarity ≈ 0.76), so aggregating them into a single physics loss remains both effective and more efficient. Decomposing the loss increases training time by roughly 30–40% due to extra gradient computations, while peak memory usage increases only marginally.

---

### Author Response · Authors · 2025-11-26
**General comment**

We want to highlight the modifications implemented in response to all reviewer feedback; the revised manuscript incorporates all these changes (highlighted in blue) in the main text and appendices I and J.

* We quantified gradient alignment using cosine similarity metrics. Specifically, we found that the conflict between physics and FM gradients is substantially reduced from approximately 20% to 7% when using ConFIG. Additionally, the gradients of the physics loss terms themselves are well aligned, with an average cosine similarity of about 0.76 for dynamic stall case.
* To address efficiency concerns, we implemented a variant that computes gradients only at the last unrolled step. This significantly reduces memory usage and training time, with minimal impact on performance.
* We conducted a detailed analysis of sampling strategies, comparing deterministic and stochastic approaches. Our results show that setting $t^* = 0.2$ consistently achieves the best trade-off between physical residuals and distributional fidelity. We also explained the origins of the observed distributional discrepancies, attributing them to optimization biases introduced during training.
* We expanded our baseline comparisons by including results for the hard-constrained ECI method.
* For hyperparameters, we provide practical guidance: to set $\sigma_\text{min}$ below the upper bound for sensitive benchmarks, use $p=1$ for residual weighting (according to noise schedule in OT-FM), $t^* = 0.2$ for stochastic sampling, and unroll 3–4 steps during training.
* We have added a clear, explicit explanation of Jensen’s gap, including a schematic figure that illustrates how gradient conflicts arise and how the ConFIG projection resolves them during unrolled training. This helps clarify the motivation and mechanism behind our approach.
* We clarified that our method is gradient-free only during inference; during training, we do differentiate through the residuals.
* Regarding loss design, we evaluated splitting the physics loss into separate terms. This increased iteration time by about 30–40% without improving accuracy, so we retained the aggregated physics loss for efficiency and simplicity.

We'd like to again thank all reviewers for their consideration!

---

### Author Response · Authors · 2025-12-02

To further assess PBFM against hard constrained methods, we added a direct comparison with PCFM on the dynamic stall case while enforcing both physical constraints; the following table reports the results.

| Metric | PBFM | OT-FM | DiffusionPDE | D-Flow | PCFM |
|---:|:---:|:---:|:---:|:---:|:---:|
| RE ·10^6 | _0.339_ | 11.02 | 12.20 | 11.32 | **0.143** |
| WD ·10^4 | **1.814** | 2.707 | _2.509_ | 3.484 | 4.013 |
| JS ·10^2 | **0.680** | _0.983_| 1.029 | 1.014 | 1.206 |
| MMSE ·10^5 | **1.490** | 2.791 | 2.626 | _2.507_ | 5.669 |
| SMSE ·10^5 | **0.874** | 1.458 | _1.236_ | 1.372 | 7.674 |
| IT [ms] | _60.47_ | **59.75** | 171.7 | 138.9 | 3906 |

The table shows that PCFM yields the lowest residual error, but this improvement comes with degraded distributional metrics (WD, JS) and substantially larger mean and standard deviation errors (MMSE, SMSE). Importantly, PCFM’s inference time is far higher (≈65× PBFM) because of its iterative correction procedure. Overall, PBFM offers a strongly improved trade-off between distributional fidelity, moment accuracy, and computational efficiency.

---

### Meta-Review · Area_Chair_VM9Y · 2025-12-15

**Summary:**

This paper introduces PBFM, a flow-matching framework for generating admissible PDE solutions. The central insight is that optimizing for distribution fidelity and physical consistency can induce conflicting gradients; the method explicitly manages this tension and targets Pareto-optimal solutions. Concretely, it combines conflict-free gradient updates with residual unrolling and stochastic sampling, and reports improved trade-offs between residual accuracy and sample quality across multiple PDE benchmarks without materially increasing inference cost. The reviewers’ main concerns appear to be satisfactorily addressed in the rebuttal, and no critical technical issues remain. Thus, I recommend acceptance.

**Reviewer Concerns:**

- Most concerns raised by Reviewer e5nq have been addressed, such as questions about whether physical residuals are sensitive to the minimum-noise hyperparameter, and the request to split the physics loss into boundary-condition and divergence-free/residual terms and report gradient-conflict metrics (e.g., cosine similarity/negative overlap) between these terms and versus the distribution-fidelity loss.

- Reviewer EdS2 raised concerns that the method lacks novelty, arguing it mainly applies an existing approach (ConFIG) to compute conflict-free gradient updates, and also requested clarification of Section 3.3. The authors responded by better articulating the claimed novelties, improving the exposition, and adding illustrative figures to clarify the approach.

- Reviewer XbPf questioned the novelty, noting that prior work has already demonstrated trade-offs between physical residual accuracy and generative accuracy.


- Reviewer f5gi raised concerns about scalability, insufficient analysis of the ODE vs. stochastic sampling discrepancy, and the need for extensive hyperparameter tuning. The authors added a variant that computes the physics residual gradient only at the final unrolling step to reduce training memory/compute while preserving residual and distributional performance, and clarified that stochastic sampling helps avoid collapse to unique solutions and improves distributional fidelity.

**Reviewer Scores:**

* Most concerns from Reviewer e5nq appear addressed; given the reviewer originally scored 10, I assume the score remains unchanged.

* Reviewer EdS2’s concerns also seem addressed; although novelty is still noted, the reviewer explicitly increased the score to 6.

* Most concerns from Reviewer XbPf may be addressed, but the novelty question (also raised by others) may remain debatable.

* Reviewer f5gi states all concerns are addressed and increased the score to 6.

---

### Decision · Program_Chairs · 2026-01-26

Accept (Poster)